# FINITE-TIME CONVERGENCE AND SAMPLE COMPLEXITY OF MULTI-AGENT ACTOR-CRITIC REINFORCEMENT LEARNING WITH AVERAGE REWARD

**Hairi and Jia Liu**
Department of Electrical and Computer Engineering
The Ohio State University
Columbus, OH 43210, USA
`hairi.1@osu.edu, liu@ece.osu.edu`

**Songtao Lu**
IBM Research AI
IBM Thomas J. Watson Research Center
Yorktown Heights, NY 10598
`songtao@ibm.com`

## ABSTRACT

In this paper, we establish the first finite-time convergence result of the actor-critic algorithm for fully decentralized multi-agent reinforcement learning (MARL) problems with average reward. In this problem, a set of $N$ agents work cooperatively to maximize the global average reward through interacting with their neighbors over a communication network. We consider a practical MARL setting, where the rewards and actions of each agent are only known to itself, and the knowledge of joint actions of the agents is not assumed. Toward this end, we propose a mini-batch Markovian sampled fully decentralized actor-critic algorithm and analyze its finite-time convergence and sample complexity. We show that the sample complexity of this algorithm is $\mathcal{O}(N^2/\epsilon^2 \log(N/\epsilon))$. Interestingly, this sample complexity bound matches that of the state-of-the-art single-agent actor-critic algorithms for reinforcement learning.

## 1 INTRODUCTION

**1) Background and Motivations:** In recent years, multi-agent reinforcement learning (MARL) has found a wide range of applications in networked large-scale systems, such as power grid systems (Riedmiller et al., 2000), autonomous driving (Yu et al., 2019; Shalev-Shwartz et al., 2016) and strategic games (Silver et al., 2018; Foerster et al., 2018), to name just a few. Although empirical successes of MARL applications have been widely observed, the fundamental theoretical understanding of how to develop fast-converging and low sample-complexity MARL algorithms, two of the most important performance metrics for MARL, remains in its infancy so far (see, e.g., (Zhang et al., 2021a) for an excellent survey). In particular, two important aspects of cooperative MARL algorithm designs deserve special attention:

- First, in the multi-agent collaborative setting, the information structure (i.e., the assumptions of who have the knowledge of what) is far more complex than its single-agent counterpart and care must be taken in MARL problem formulations. In the cooperative MARL literature so far, many existing works assume full knowledge of joint states and joint actions, which often do not hold true in practice. For example, in autonomous driving (Yu et al., 2019), each vehicle can only observe/detect the actions of the surrounding vehicles that are within its communication range. As another example, in power grid networks (Riedmiller et al., 2000), each power distributor generally does not know the resistor values set by other distributors.

- Second, most MARL theoretical studies in the MARL literature are focused on the discounted total reward setting, where a hyperparameter $\gamma \in (0, 1)$ is introduced as the discount factor in the objective function. Although the discounted total reward setting captures the important aspect of diminishing return in the future, it may not be appropriate for many other applications where the *long-term average reward* is of interest. For example, in the optimization of distributed communication networks with MARL, the typical and natural performance metrics are long-term average throughput or latency in the steady-state.

The lack of a fundamental understanding on how to develop efficient cooperative MARL algorithms that consider the above two aspects in terms of information structure, scalability, and communication

and sample complexities motivates us to fill this gap by developing a fully decentralized cooperative MARL algorithm in the average reward setting, without assuming joint action knowledge.

**2) Technical Challenges:** Developing a fully decentralized cooperative MARL algorithm for the average reward setting without full joint action knowledge is highly non-trivial and several major technical challenges naturally arise. First, it is well-known that, even in the single-agent reinforcement learning (RL) setting, the average reward setting is more challenging to analyze compared to the discounted reward setting, which necessitates different proof techniques (Tsitsiklis & Van Roy, 1999). In MARL, the decentralized nature and the lack of joint action-state information further complicate the algorithm design and analysis in the average reward setting. Second, due to the lack of joint action knowledge, the communication costs among agents in MARL will be significantly increased to achieve a satisfying performance, which implies that low sample and communication complexities are even more challenging and critical for MARL without joint action information.

**3) Main Results and Contributions:** The main contribution of this paper is that we overcome the aforementioned challenges and develop a consensus-based Markovian sampled decentralized actor-critic algorithm for MARL. Our key results in this paper are summarized as follows:

- We propose a batch-sampled actor-critic algorithm that uses consensus updates in TD-sharing among agents. This batch sampling approach enables more efficient communication compared to the classical fully decentralized MARL. Specifically, in order to converge to an $\epsilon$ neighborhood of the stationary, we require $\mathcal{O}(\epsilon^{-1}\log(\epsilon^{-1}))$ rounds of communication while only needing $\mathcal{O}(\epsilon^{-2}\log(\epsilon^{-1}))$ samples. By contrast, the state-of-the-art MARL requires a communication round per sampling. Also, our algorithm allows the use of constant step-sizes in both the actor and the critic steps.

- We provide the first-ever sample complexity analysis in the MARL average reward problem setting without joint action information. Our obtained complexity is $\mathcal{O}(\epsilon^{-2}\log(\epsilon^{-1}))$, where $\epsilon$ is the closeness to the neighborhood of stationary point (treating network size $N$ as a fixed constant). It is worth noting that the order-wise sample complexity of our algorithm matches that of the state-of-the-art single-agent RL algorithms.

## 2 RELATED WORK

In this section, we provide a quick overview on the closely related work on MARL algorithms and their theoretical results, along with several notable related counterparts in single-agent RL.

**1) MARL Theoretical Analysis and Algorithm Design:** For recent advances in MARL algorithms and their theoretical results, Zhang et al. (2021a) provided a comprehensive survey. Also, Lee et al. (2020) highlighted the evolution from single-agent to multi-agent RL from a distributed optimization perspective. In the broader area of MARL, a line of research has been focused on the MARL policy evaluation problem. These works analyzed the convergence Doan et al. (2019) and proposed various variance reduction of policy evaluation in decentralized MARL algorithms Zhang et al. (2021b). Doan et al. (2019) used i.i.d. sampling and has shown the sample complexity of $\tilde{\mathcal{O}}(\epsilon^{-1})$ for their TD(0) learning algorithm to reach a mean-square error convergence. However, these algorithms do not involve policy improvement and solely focus on the performance evaluation of given policies. In the areas of joint policy evaluation and improvement, Foerster et al. (2018) considered multi-agent actor-critic algorithm that has a centralized critic and decentralized actors, which is different from our fully decentralized actor-critic algorithm. In contrast, Zhang et al. (2018) established asymptotic convergence results for fully decentralized MARL actor-critic algorithms. Concurrent with our work, Chen et al. (2021) has recently studied the mini-batch Markovian sampling actor-critic algorithm for a class of discounted reward MARL problems, where the finite-time convergence result is obtained. They have applied batch sampling for both actor and critic steps and achieved a sample complexity of $O(\epsilon^{-2}\log(\epsilon^{-1}))$, which is the same as ours. We note that, together with our work, these are the first finite-time convergence results for MARL. However, there are several key differences between our work and (Chen et al., 2021). First, we focus on the average reward problem, while Chen et al. (2021) studied the discounted reward setting. Second, in (Chen et al., 2021), agents share a noisy version of the rewards with the neighbors, which requires a re-sampling process from every sampled reward instance. In contrast, we allow agents to share local TD-errors with their neighbors and no re-sampling is required.

Table 1: Comparison of sample complexity of single-agent (SA) and multi-agent (MA) AC algorithms and TD(0) algorithms at Average Reward (AR) and Discounted Reward (DR) settings.

| Paper | Problem | Sampling | | Sample Complexity |
|---|---|---|---|---|
| | | actor step | critic step | |
| Qiu et al. (2021) | SAAR | i.i.d. | Markovian | $\mathcal{O}(\epsilon^{-3}\log^2(\epsilon^{-1}))$ |
| Xu et al. (2020) | SADR | Markovian | Markovian | $\mathcal{O}(\epsilon^{-2}\log(\epsilon^{-1}))$ |
| Zhang et al. (2018) | MAAR | Markovian | Markovian | Asymptotic |
| Doan et al. (2019) | MADR | N/A | i.i.d | $\tilde{\mathcal{O}}(\epsilon^{-1})$ |
| Chen et al. (2021) | MADR | Markovian | Markovian | $\mathcal{O}(\epsilon^{-2}\log(\epsilon^{-1}))$ |
| **This paper** | **MAAR** | Markovian | Markovian | $\mathcal{O}(N^2/\epsilon^2\log(N/\epsilon))$ |

**2) Related Literature in Single-Agent RL:** We note that single-agent RL can be viewed as a centralized approach, where a central controller collects joint actions, rewards and even designs policies for agents. For the single-agent average reward setting, Tsitsiklis & Van Roy (1999) first analyzed the asymptotic convergence of TD($\lambda$) algorithm with function approximations in the policy evaluation problem. Also, Tsitsiklis & Van Roy (2002) provided insights in terms of differences and connections between average reward and discounted reward of TD-based learning algorithms with function approximations. Recently, Qiu et al. (2021) analyzed the sample complexity for an actor-critic algorithm for the average reward problem. In their actor-critic algorithm, they used batch sampling for the critic learning and i.i.d. sampling for the actor step, with sample complexity being $\mathcal{O}(\epsilon^{-3}\log^2(\epsilon^{-1}))$. By applying mini-batch sampling update, we are able to improve the sample complexity by a factor of $\mathcal{O}(\epsilon^{-1})$. Another closely related work on single-agent RL is (Xu et al., 2020), where the authors studied the discounted reward problem. They used batch sampling for both actor and critic steps in their actor-critic algorithm and developed a new technique to handle bias error in the critic step, which we also adopted for the average approximation parameter analysis in our critic step. This achieved the state-of-the-art sample complexity of $\mathcal{O}(\epsilon^{-2}\log(\epsilon^{-1}))$ for single-agent RL. In addition, the global convergence of actor-critic algorithm to the optimal policy has been studied in the case of discounted setting with single time scale in (Fu et al., 2020) and linear quadratic regulator in (Yang et al., 2019). However, we note that these settings are fundamentally different from the average reward setting and it will be an interesting future direction to consider global convergence possibility in the average reward setting.

To conclude this section, we summarize the aforementioned related actor-critic and TD algorithms and their sample complexity results in Table 1.

## 3 MULTI-AGENT REINFORCEMENT LEARNING WITH AVERAGE REWARD

### 3.1 SYSTEM MODEL

Consider a multi-agent system with $N$ agents, denoted by $\mathcal{N} = \{1, \cdots, N\}$, operating in a networked environment. Let $\mathcal{E}$ be the edge set for a given network $\mathcal{G} = (\mathcal{N}, \mathcal{E})$. To formulate our MARL problem and facilitate our subsequent discussions, we first define the notion of networked multi-agent MDP as follows.

**Definition 1** (Networked Multi-Agent MDP). Let $\mathcal{G} = (\mathcal{N}, \mathcal{E})$ be a communication network that connects $N$ agents. A networked multi-agent MDP is defined by following tuple $(\mathcal{S}, \{\mathcal{A}^i\}_{i \in \mathcal{N}}, P, \{R^i\}_{i \in \mathcal{N}}, \mathcal{G})$, where $\mathcal{S}$ is the global state space observed by all agents, $\mathcal{A}^i$ is the action set for agent $i$, and $P : \mathcal{S} \times \mathcal{A} \times \mathcal{S} \to [0, 1]$ is a global state transition function, and $R^i : \mathcal{S} \times \mathcal{A}$ is the local reward function for agent $i$. Let $\mathcal{A} = \prod_{i \in \mathcal{N}} \mathcal{A}^i$ be the joint action set of all agents.

In this paper, we assume that the global state space $\mathcal{S}$ and action space for agent $\mathcal{A}^i$ are finite. As a result, the joint action space $\mathcal{A}$ is also finite for finite $N$. We also note that at time $t \geq 0$, all agents can observe the current global state $s_t$. However, agent $i$ can *only* observe its own action $a_t^i \in \mathcal{A}^i$, which is the key difference between our model and that in (Zhang et al., 2018), where it is assumed that the joint actions are observable to all agents. Moreover, each agent can only observe its own reward $r_t^i$, i.e., agents do not observe or share rewards with other agents at time $t$. The

reward function $R^i(s, a)$ is an expectation given $s$ and $a$, and the instantaneous reward is denoted by $r^i(s, a)$, i.e., $R^i(s, a) = \mathbb{E}[r^i(s, a)]$.

We consider policies that are stationary. In our MARL system, each agent chooses its action following its local policy $\pi^i$ that is conditioned on the current global state $s$, i.e., $\pi^i(a^i|s)$ is the probability for agent $i$ to choose an action $a^i \in \mathcal{A}^i$. Then, the joint policy $\pi : \mathcal{S} \times \mathcal{A} \to [0, 1]$ can be written as $\pi(a|s) = \prod_{i \in \mathcal{N}} \pi^i(a^i|s)$.

Moreover, the policies at the agents are parameterized. Specifically, each agent $i$'s local policy can be written as $\pi_{\theta^i}^i$, where $\theta^i \in \mathbb{R}^{m_i}$ denotes the parameter. We let $\theta \triangleq [(\theta^1)^T, \cdots, (\theta^N)^T]^T \in \mathbb{R}^{\sum_{i=1}^N m_i}$. Then, we can write the joint policy as follows: $\pi_\theta(a|s) = \prod_{i \in \mathcal{N}} \pi_{\theta^i}(a^i|s)$.

## 3.2 Technical Assumptions

We now state the following assumptions on the positivity and continuity of $\pi_{\theta^i}^i(a^i|s)$, which guarantee the stationary distribution of $\{s_t\}$ under any given policy.

**Assumption 1.** For any $i \in \mathcal{N}$, $s \in \mathcal{S}$, $a^i \in \mathcal{A}^i$ and $\theta^i \in \mathbb{R}^{m_i}$, the policy function $\pi_{\theta^i}^i(a^i|s) \geq 0$. Also, $\pi_{\theta^i}^i(s, a)$ is a continuously differentiable with respect to the parameter $\theta^i$. In addition, for any $\theta$, we assume the induced Markov chain $\{s_t\}_{t \geq 0}$ is irreducible and aperiodic, and its transition matrix $P^\theta$ is $P^\theta(s'|s) = \sum_{a \in \mathcal{A}} \pi_\theta(a|s) \cdot P(s'|s, a)$, $\qquad \forall s, s' \in \mathcal{S}$.

Assumption 1 guarantees that the states have a stationary distribution $d_\theta(s)$ over $\mathcal{S}$ given any policy $\pi_\theta$. As a result, the Markov chain of state action pair $\{(s_t, a_t)\}$ also has a stationary distribution $d_\theta(s) \cdot \pi_\theta(a|s)$.

**Assumption 2.** The instantaneous reward $r_t^i$ is uniformly bounded by a constant $r_{\max} > 0$ for any $i \in \mathcal{N}$ and $t \geq 0$.

Assumption 2 is common in the literature (see, e.g., (Zhang et al., 2018; Xu et al., 2020; Doan et al., 2019)) and easy to be satisfied in many practical MDP models with finite state and action spaces.

**Assumption 3.** Let $A$ be a consensus weight matrix for a given communication network $\mathcal{G}$. There exists a positive constant $\eta > 0$ such that $A \in \mathbb{R}^{N \times N}$ is doubly stochastic and $A_{ii} \geq \eta$, $\forall i \in \mathcal{N}$. Moreover, $A_{ij} \geq \eta$ if $i, j$ are connected, otherwise $A_{ij} = 0$ for all $i, j$.

Assumption 3 is standard in the distributed multi-agent optimization literature Nedic & Ozdaglar (2009). We remark that for a practical choice of $A$, one can use the following form $A = \frac{1}{\deg(\mathcal{G})}(\deg(\mathcal{G}) \cdot I - L)$, where $\deg(\mathcal{G})$ is the degree of the graph $\mathcal{G}$ (i.e. the maximal vertex degree), $I$ is the identity matrix of conforming dimensionality, and $L$ is the Laplacian matrix of the graph. It is easy to verify that this matrix is symmetric, doubly stochastic and $\eta \geq \frac{1}{\deg(\mathcal{G})} \geq \frac{1}{N}$.

**Assumption 4.** Each agent $i$'s value function is parameterized by the class of linear functions, i.e., $V_\theta(s; w) = \phi(s)^T w$ where $\phi(s) \triangleq [\phi_1(s), \cdots, \phi_K(s)]^T \in \mathbb{R}^K$ is the feature associated with the state $s \in \mathcal{S}$ and $K < |\mathcal{S}|$. The feature vectors $\phi(s)$ are uniformly bounded for any $s \in \mathcal{S}$. Without loss of generality, we assume that $\|\phi(s)\| \leq 1$. Furthermore, the feature matrix $\Phi \in \mathbb{R}^{|\mathcal{S}| \times K}$ has full column rank. Also, for any $u \in \mathbb{R}^K$, $\Phi u \neq \mathbf{1}$, where $\mathbf{1}$ is an all-one vector.

This assumption on features is standard and has been widely adopted in the literature, e.g., (Tsitsiklis & Van Roy, 1999; Zhang et al., 2018; Qiu et al., 2021). This assumption implies the following property: for any policy $\pi_\theta$, the inequality $w^T A_{\pi_\theta} w < 0$ holds for any $w \neq 0$, where $A_{\pi_\theta}$ is defined as

$$A_{\pi_\theta} := \mathbb{E}_{s \sim d_\theta(s), s' \sim P(\cdot|s)}[(\phi(s') - \phi(s))\phi^T(s)]. \tag{1}$$

This property further implies that for all $\theta$, $A_{\pi_\theta}$ is invertible and $\lambda_{\max}(A_{\pi_\theta} + A_{\pi_\theta}^T) < 0$ (Qiu et al., 2021), where $\lambda_{\max}(\cdot)$ is the largest eigenvalues of the matrix.

**Assumption 5.** There exists a constant $\lambda_A > 0$ such that $\lambda_{\max}(A_{\pi_\theta} + A_{\pi_\theta}^T) \leq -\lambda_A$ holds for all $\theta \in \mathbb{R}^{\sum_{i \in \mathcal{N}} m_i}$.

Assumption 5 ensures the optimal approximation $w_\theta^*$ for any given policy $\pi_\theta$ is uniformly bounded (see discussion before Theorem 2 and Lemma 4).

**Assumption 6.** Let $\psi_\theta(s, a) = \nabla_\theta \log \pi_\theta(a|s)$ be the score function for any state-action pair $(s, a)$. For any two policy parameters $\theta, \theta' \in \mathbb{R}^{\sum_{i \in \mathcal{N}} m_i}$, and any state-action pair $(s, a) \in \mathcal{S} \times \mathcal{A}$, there exist positive constants such that the following hold: 1): $\|\psi_\theta(s, a)\| \le C_\psi$; 2): $\|\nabla_\theta J(\theta) - \nabla_\theta J(\theta')\| \le L_J \|\theta - \theta'\|$; where $J(\theta)$ is defined in (2) and $\|\cdot\|$ denotes the $\ell_2$-norm.

Assumption 6 says that the score function is uniformly bounded for any policy and the gradient of the objective function has a Lipschitz property with respect to the policy parameter. This assumption has also been adopted in the analysis of the single-agent actor-critic algorithm in (Qiu et al., 2021). We note that for the discounted reward problem, this gradient Lipschitz property can be guaranteed through (Xu et al., 2020, Assumption 2). We note that Assumption 6 can be satisfied by the class of soft-max policy under the Assumption 1, as in (Guo et al., 2021).

### 3.3 THE OBJECTIVE FUNCTION

The goal of the agents is to find a joint policy $\pi_\theta$ to maximize the global average long-term reward. Mathematically, this can be written as:

$$\text{maximize}_\theta \qquad J(\theta) = \lim_{T \to \infty} \frac{1}{T} \mathbb{E} \left( \sum_{t=0}^{T-1} \frac{1}{N} \sum_{i \in \mathcal{N}} r_{t+1}^i \right) = \sum_{s \in \mathcal{S}} d_\theta(s) \sum_{a \in \mathcal{A}} \pi_\theta(a|s) \cdot \bar{R}(s, a), \quad (2)$$

where $\bar{R}(s, a) = \frac{1}{N} \sum_{i \in \mathcal{N}} R^i(s, a)$ is the global average reward function. Let $\bar{r}_t = \frac{1}{N} \sum_{i \in \mathcal{N}} r_t^i$, then we have $\bar{R}(s, a) = \mathbb{E}[\bar{r}_{t+1}|s_t = s, a_t = a]$. Next, we define the state-action value function: $Q_\theta(s, a) = \mathbb{E}[\sum_{t=0}^\infty \bar{r}_{t+1} - J(\theta)|s_0 = s, a_0 = a, \pi_\theta]$, and the state value function $V_\theta(s) = \sum_{a \in \mathcal{A}} Q_\theta(s, a) \cdot \pi_\theta(a|s)$. The advantage function is defined as follows:

$$\text{Adv}_\theta(s, a) = Q_\theta(s, a) - V_\theta(s). \quad (3)$$

### 3.4 POLICY GRADIENT THEOREM

The gradient of a policy $\pi_\theta$ for decentralized policy gradient is stated in the following theorem.

**Theorem 1** (Policy Gradient Theorem for MARL (Zhang et al., 2018)). *For any $\theta$, let $\pi_\theta : \mathcal{S} \times \mathcal{A} \to [0, 1]$ be a policy and let $J(\theta)$ be the global average long-term average return defined in (2). Then, the gradient of $J(\theta)$ with respect to parameter $\theta^i$ can be computed as:*

$$\nabla_{\theta^i} J(\theta) = \mathbb{E}_{s \sim d_\theta, a \sim \pi_\theta} [\nabla_{\theta^i} \log \pi_{\theta^i}^i(a^i|s) \cdot \text{Adv}_\theta(s, a)]. \quad (4)$$

## 4 A CONSENSUS-BASED ACTOR-CRITIC ALGORITHM

In this section, we propose a consensus-based actor-critic algorithm that includes two key steps: actor and critic. In the critic step, the algorithm evaluates the value functions for the policy $\pi_{\theta_t}$ at time $t$. After the critic step, the algorithm enters the actor step, which improves the policy parameter $\theta_t$ according to the direction from policy gradient as shown in Theorem 1. In both steps, we use constant step-sizes and adopt batch sampling.

In this paper, we use linear function approximations for the value functions. Specifically, each agent $i$ has a parameter $w^i \in \mathbb{R}^K$ to approximate the global value functions $V_\theta(s; w^i)$ for each state $s \in \mathcal{S}$. For linear approximation, we have $V_\theta(s; w^i) = \phi(s)^T w^i$, where $\phi(s) \in \mathbb{R}^K$ denotes the feature for state $s$. As a result, the gradient of value function at state $s$ with respect to approximation parameter $w^i$ is $\phi(s)$, i.e. $\nabla_w V_\theta(s; w^i) = \phi(s)$.

**1) The Critic Step:** The critic step is achieved through its own oracle, which is summarized in Algorithm 1. In the critic step, we allow the agents to communicate the approximation parameters $w(s)$ with their neighbors via the communication network with consensus weight matrix $A$. For agent $i \in \mathcal{N}$, the parameter is locally updated by following rules:

$$\mu_{k,\tau+1}^i = (1 - \beta)\mu_{k,\tau}^i + \beta r_{k,\tau+1}^i \quad (5)$$

$$\delta_{k,\tau}^i = r_{k,\tau+1}^i - \mu_{k,\tau}^i + \phi^T(s_{k,\tau+1})w_k^i - \phi^T(s_{k,\tau})w_k^i \quad (6)$$

---

**Algorithm 1:**      **Mini-batch TD learning for Critic**

---

**Input** : $s_0, \pi_{\theta_t}, \phi$, step-size $\beta$, critic step iteration number $K$, critic batch size $M$, the communication network $A$

1 **for** $k = 0, \cdots, K_c - 1$ **do**
2      $s_{k,0} = s_{k-1,M}$ ( when $k = 0$, $s_{k,0} = s_0$);
3      **for** *all* $i \in \mathcal{N}$ **do**
4          **for** $\tau = 0, \cdots, M - 1$ **do**
5              Execute action $a^i_{k,\tau} \sim \pi^i_{\theta^i_t}(\cdot|s_{k,\tau})$;
6              Observe the state $s_{k,\tau+1}$ and reward $r^i_{k,\tau+1}$;
7              Update $\mu^i_{k,\tau+1} \leftarrow (1-\beta) \cdot \mu^i_{k,\tau} + \beta \cdot r^i_{k,\tau+1}$;
8              Update $\delta^i_{k,\tau} \leftarrow r^i_{k,\tau+1} - \mu^i_{k,\tau} + \phi^T(s_{k,\tau+1})w^i_k - \phi^T(s_{k,\tau})w^i_k$;
9          **end**
10          **Critic Step:** $\tilde{w}^i_k \leftarrow w^i_k + \frac{\beta}{M}\sum_{\tau=0}^{M-1} \delta^i_{k,\tau} \cdot \phi(s_{k,\tau})$;
11          **Consensus Update** $w^i_{k+1} \leftarrow \sum_{j \in \mathcal{N}_i} A(i,j) \cdot \tilde{w}^j_k$;
12      **end**
13 **end**

**Output:** $s_{K_c-1,M}, \quad w_{K_c}$

---

$$\tilde{w}^i_k = w^i_k + \frac{\beta}{M}\sum_{\tau=0}^{M-1} \delta^i_{k,\tau} \cdot \phi(s_{k,\tau}), \tag{7}$$

where $\beta > 0$ is the step-size of the critic step, $\mu^i_{k,\tau}$ is the estimate of the long-term return of agent $i$, and $\delta^i_{k,\tau}$ is the local TD-error for agent $i$ at iteration $k$ using sample $\tau$. Here, in each iteration $k$ in the critic step, the approximation parameter is locally updated through a batch of sampling as in (7), where the batch size is $M$. Then, agent $i$ will further update the approximation parameter $w^i$ through a weighted average of its local and neighboring agents' parameters as follows: $w^i_{k+1} = \sum_{j \in \mathcal{N}_i} A(i,j)\tilde{w}^i_k$. This batched sampling update continues for $K_c$ iterations for each given policy $\pi_{\theta_t}$.

**2) The Actor Step:** As shown in Theorem 1, the advantage function needs to be known to compute the gradient. However, from the definition in (3), the joint action $a$ also has to be known to compute the advantage function, whereas in our model, each agent can only observe its own action. As a result, an estimation of the advantage function is required. Here, we show that the global TD-error is an unbiased estimate of the advantage function.

At time $t$, suppose we have samples $s_t, a_t, s_{t+1}$ and the rewards $\{r^i_{t+1}\}_{i \in \mathcal{N}}$ then the advantage function is as follows: $\text{Adv}_\theta(s_t, a_t) = \mathbb{E}[\bar{r}_{t+1} - J(\theta) + V_\theta(S_{t+1}) - V_\theta(s_t)|s_t, a_t]$, and the global TD-error can be computed as follows: $\delta_t = \bar{r}_{t+1} - \mu_t + V_\theta(s_{t+1}) - V_\theta(s_t)$, where $\mu_{t+1} = (1-\alpha) \cdot \mu_t + \alpha \cdot \bar{r}_{t+1}$ is the estimate for the average long term return, and $\alpha$ is the step-size for the actor step. Hence, we have that the expected global TD-error is the advantage function, i.e., $\mathbb{E}[\delta_t|s_t, a_t] = A_\theta(s_t, a_t)$. Thus, we can use this global TD-error as an unbiased estimate of the advantage function. For agent $i$, the local TD-error can be computed as $\delta^i_t = r^i_{t+1} - \mu^i_t + V_\theta(s_{t+1}) - V_\theta(s_t)$, where $\mu^i_{t+1} = (1-\alpha)\mu^i_t + \alpha r^i_{t+1}$. We also note that $\mu_t = \frac{1}{N}\sum_{i \in \mathcal{N}} \mu^i_t$.

Thus, once each agent knows the global TD-error, the policy parameter can be updated according to the policy gradient rule in (4). However, without any communication, each agent only has the knowledge of its own local TD-error. Moreover, we will show that the networked TD-error is actually the average of the local TD-errors. Specifically,

$$\delta_t = \bar{r}_{t+1} - \mu_t + V_\theta(s_{t+1}) - V_\theta(s_t) = \frac{1}{N}\sum_{i \in \mathcal{N}} r^i_{t+1} - \frac{1}{N}\sum_{i \in \mathcal{N}} \mu^i_t + V_\theta(s_{t+1}) - V_\theta(s_t)$$

$$= \frac{1}{N}\sum_{i \in \mathcal{N}}[r^i_{t+1} - \mu^i_t + V_\theta(s_{t+1}) - V_\theta(s_t)] = \frac{1}{N}\sum_{i \in \mathcal{N}} \delta^i_t.$$

---

**Algorithm 2:**     **Minibatch-TD sharing for Actor Critic Algorithm**

---

**Input** : state feature matrix $\Phi$, actor step-size $\alpha$, Initial parameters $\theta_i$ for all $i \in \mathcal{N}$

1   **for** $t = 0, \cdots, T - 1$ **do**
2      **critic update:** $w_t, s_{t,0} = $ Minibatch-TD-critic in Algorithm 1;
3      **for** $l = 0, \cdots, B - 1$ **do**
4          **for** *all* $i \in \mathcal{N}$ **do**
5              Execute action $a_{t,l}^i \sim \pi_{\theta_t^i}^i(\cdot|s_{t,l})$;
6              Observe the state $s_{t,l+1}$ and reward $r_{t,l+1}^i$;
7              Update $\mu_{t,l+1}^i \leftarrow (1 - \alpha) \cdot \mu_{t,l}^i + \alpha \cdot r_{t,l+1}^i$;
8              Update $\delta_{t,l}^i \leftarrow r_{t,l+1}^i - \mu_{t,l}^i + \phi^T(s_{t,l+1})w_t^i - \phi^T(s_{t,l})w_t^i$;
9              Update $\psi_{t,l}^i \leftarrow \nabla_{\theta^i} \log \pi_{\theta_t^i}^i(s_{t,l}, a_{t,l}^i)$;
10          **end**
11      **end**
12      Let $\Delta_0 = \begin{bmatrix} \delta_{t,0}^1 & \cdots & \delta_{t,0}^N \\ \vdots & \ddots & \vdots \\ \delta_{t,B-1}^1 & \cdots & \delta_{t,B-1}^N \end{bmatrix}$;
13      **for** $i \in \mathcal{N}$ **do**
14          **for** $k = 0 : t_{gossip} - 1$ **do**
15              $\Delta_{k+1}(:, i) \leftarrow \sum_{j \in \mathcal{N}_i} A(i, j) \cdot \Delta_k(:, j)$;
16          **end**
17      **end**
18      **for** *all* $i \in \mathcal{N}$ **do**
19          Let $\tilde{\delta}_{t,1:B-1}^i = \Delta_{t_{gossip}}(:, i)$ ;
20          **Actor Step:** $\theta_{t+1}^i \leftarrow \theta_t^i + \frac{\alpha}{B} \sum_{l=0}^{B-1} \tilde{\delta}_{t,l}^i \cdot \psi_{t,l}^i$;
21      **end**
22 **end**

**Output:** $\theta_{\hat{T}}$ with $\hat{T}$ chosen uniformly from $\{1, \cdots, T\}$

---

For any time $t$, the average of the local TD-errors is an unbiased estimate of the advantage function. Therefore, we just need to let each agent communicate with its neighbors so that an average of all local TD-errors can be reached or estimated for all agents.

From the results in (Nedic & Ozdaglar, 2009), we have $\lim_{\tau \to \infty} A^\tau(x^1, \cdots, x^N)^T = \frac{1}{N}\sum_{i \in \mathcal{N}} x^i \mathbf{1}$. However, this convergence is asymptotic, meaning that the exact estimation can only be achieved with infinite iterations (i.e., $\tau \to \infty$). In practice, since one can only apply finite iterations, we use $\tilde{\delta}_t^i$ to denote the estimate of the global TD-error maintained by agent $i$ after $t_{gossip}$ iterations of updates at time $t$, i.e., $\tilde{\delta}_t^i = [A^{t_{gossip}}]_i \Delta_t^0$, where $\Delta_t^0 = (\delta_t^1, \cdots, \delta_t^N)^T$ is the $N$-dimension vector of local TD-errors at time $t$. We note that agents do not need to know the weight information of other agents. Rather, each agent just needs to exchange updated estimate of the local TD-errors with its neighbors for $t_{gossip}$ rounds as shown in Lines 11-16 of Algorithm 2. This communication among agents is also done in a batch fashion, with batch size being $B$. This implies that for each outer iteration $t \in \{0 \cdots, T - 1\}$, only $t_{gossip}$ rounds of communication for every $B$ samples are needed.

Combined with a $B$-batched Markovian sampling, the parameter $\theta^i$ update for agent $i \in \mathcal{N}$ can be written as follows: $\theta_{t+1}^i = \theta_t^i + \frac{\alpha}{B} \sum_{l=0}^{B-1} \tilde{\delta}_{t,l}^i \cdot \psi_{t,l}^i$ where $\alpha$ is the step-size for the actor step and $\psi_{t,l}^i = \nabla_{\theta^i} \log \pi_{\theta_t^i}^i(s_{t,l}, a_{t,l}^i)$ is the local score function for agent $i$ using $l$-th sample at time $t$. The actor step of our algorithm is illustrated in Algorithm 2.

# 5 THEORETICAL CONVERGENCE ANALYSIS

In this section, we present the convergence results for both the critic and actor steps in Theorems 2 and 3, respectively. Due to space limitation, we relegate the proofs to the supplementary material.

## 5.1 CONVERGENCE ANALYSIS FOR CRITIC (ALGORITHM 1)

For a given policy $\pi_\theta$, we define $b_{\pi_\theta} = \mathbb{E}_{s\sim d_\theta, a\sim\pi_\theta}[\phi(s)(\bar{r}(s,a) - J(\theta))]$, where $\bar{r}(s,a) = \frac{1}{N}\sum_{i\in\mathcal{N}} r^i(s,a)$ and $J(\theta)$ are as defined in (2). For all agents, the optimal solution (Wu et al., 2020), (Qiu et al., 2021) of this critic learning is $w_\theta^* = -A_{\pi_\theta}^{-1} b_{\pi_\theta}$, where $A_{\pi_\theta}$ is defined as in (1). The invertiblity of $A_{\pi_\theta}$ is due to the Assumptions 1 and 4 (see more details in (Qiu et al., 2021; Tsitsiklis & Van Roy, 1999)). Then, the convergence of the critic step is summarized as follows:

**Theorem 2** (Convergence and Sample Complexity of the Critic). *Suppose that Assumptions 1-5 hold. For any given policy $\pi_\theta$, consider the iteration generated by Algorithm 1. Recall the definition of $\lambda_A$ in the Assumption 5 and let $\beta < \min\{\frac{\eta^{N-1}}{2(1-\eta^{N-1})}, \frac{\lambda_A}{128}, \frac{4}{\lambda_A}\}$. It then follows that:*

$$\mathbb{E}[\sum_{i=1}^N ||w_{K_c}^i - w_\theta^*||^2] \leq \kappa_1' N^4 \gamma^{2K_c} + \kappa_2'' N^6 \beta^2 + \kappa_3 N^5 \gamma^{K_c}\beta$$

$$+ 2N(1 - \frac{\lambda_A}{8}\beta)^{K_c}||\bar{w}_0 - w_\theta^*||_2^2 + \frac{\kappa_4}{M}N, \qquad (8)$$

*where $\gamma := (1-\eta^{N-1})\cdot(1+2\beta) < 1$ and $\kappa_1', \kappa_2'', \kappa_3, \kappa_4$ are positive constants. If we further let $K_c \geq \frac{1}{2}\max\{\log_{\gamma^{-1}}\frac{6\kappa_1' N^4}{\epsilon}, \log_{(1-\frac{\lambda_A}{8}\beta)^{-1}}\frac{12N||\bar{w}_0 - w_\theta^*||_2^2}{\epsilon}\}$, $\beta \leq \min\{\frac{1}{N^3}\sqrt{\frac{\epsilon}{6\kappa_2''}}, \frac{\eta^{N-1}}{2(1-\eta^{N-1})}, \frac{\lambda_A}{128}, \frac{4}{\lambda_A}\}$ and $M \geq \frac{6\kappa_4 N}{\epsilon}$, then we have $\mathbb{E}[\sum_{i=1}^N ||w_{K_c}^i - w_\theta^*||^2] \leq \epsilon$ for all $i \in \mathcal{N}$ with total sample complexity given by $K_c M = \mathcal{O}(\frac{N}{\epsilon}\log(\frac{N}{\epsilon}))$.*

Theorem 2 establishes a convergence result for the policy evaluation of a given policy $\pi_\theta$. We can see that our constant step-size batch-sampled critic process can achieve the same complexity of $\tilde{\mathcal{O}}(\epsilon^{-1})$ order-wise as the TD(0) learning in Doan et al. (2019), where diminishing step-sizes were used. On the other hand, in the single agent average reward setting of Qiu et al. (2021), there exists a non-vanishing error term in their critic convergence. In contrast, with proper choices of iteration number $K_c$ and batch size $M$, our mean-square error can be arbitrarily small.

## 5.2 CONVERGENCE ANALYSIS FOR ACTOR-CRITIC ALGORITHM (ALGORITHM 2)

Define the approximation error introduced by using linear approximation in the critic step, $\xi_{\text{approx}}^{\text{critic}} = \max_{\theta\in\mathbb{R}^{\Sigma_{i\in\mathcal{N}} m^i}}\mathbb{E}_{s\sim d_\theta}[|V_{\pi_\theta}(s) - V_{w_{\pi_\theta}^*}(s)|^2]$. For a given policy $\pi_\theta$, this error represents the gap between ground truth value function under such policy and the value function obtained by the best possible linear approximation. Such an error term is standard in the literature where linear approximations are adopted (Qiu et al., 2021; Xu et al., 2020). Let $R_w$, defined in Lemma 4, be an upper bound on $||w_\theta||$ for all policy parameter $\theta$.

**Theorem 3** (Overall Convergence Rate and Sample Complexity). *Suppose that Assumption 1-6 hold. Consider the actor-critic algorithm in Algorithm 2. Let step-size $\alpha = \frac{1}{4L_J}$. It then holds that:*

$$\mathbb{E}[||\nabla_\theta J(\theta_{\tilde{T}})||^2] \leq \frac{16L_J r_{\max}}{T} + 72N\frac{\sum_{t=1}^T \sum_{i=1}^N ||w_t^i - w_{\theta_t}^*||^2}{T} + 18\kappa_3 N^3(1-\eta^{N-1})^{2t_{gossip}}$$

$$+ 72\xi_{approx}^{critic} + 288\frac{(r_{\max} + R_w)^2[1 + (2\kappa - 1)\rho]}{B(1-\rho)}N, \qquad (9)$$

*where $\kappa_3$ is a positive constant. Furthermore, let $T \geq \frac{64L_J r_{\max}}{\epsilon}$, $B \geq 576\frac{(r_{\max}+2R_w)^2[1+(2\kappa-1)\rho]}{(1-\rho)}\frac{N}{\epsilon}$ and the communication round among the agents $t_{gossip} \geq \frac{1}{2}\log_{(1-\eta^{N-1})^{-1}}\frac{64\kappa_3 N^3}{\epsilon}$. Suppose for the same setting as in Theorem 1 holds so that $\mathbb{E}[\sum_{i=1}^N ||w_{K_c}^i - w_\theta^*||^2] \leq \frac{\epsilon}{288N}$ for all $0 \leq t \leq T$, then we have $\mathbb{E}[||\nabla_\theta J(\theta_t)||^2] \leq \epsilon + \mathcal{O}(\xi_{approx}^{critic})$, with a total complexity of $(B + MK_c)T = \mathcal{O}(\frac{N^2}{\epsilon^2}\log(\frac{N}{\epsilon}))$. And the communication complexity is $(K_c + t_{gossip})T = \mathcal{O}(\frac{1}{\epsilon}\log(\frac{N}{\epsilon}))$.*

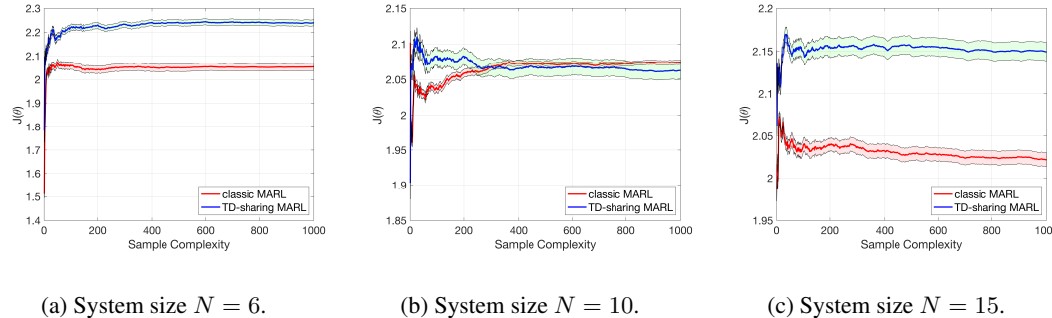

(a) System size $N = 6$.      (b) System size $N = 10$.      (c) System size $N = 15$.

Figure 1: Our TD-sharing algorithm vs classical MARL algorithm.

Theorem 3 concludes the overall sample complexity of our proposed actor-critic algorithm. The sample complexity of $\mathcal{O}(\epsilon^{-2} \log(\epsilon^{-1}))$ matches the state-of-the-art single-agent actor-critic RL by Xu et al. (2020) and the discounted MARL by Chen et al. (2021).

We note that the overall communication complexity also matches that of (Chen et al., 2021) in the discounted reward setting. However, our work is still an improvement compared to the classical MARL in Zhang et al. (2018) for the average reward setting. Specifically, Zhang et al. (2018) needed a communication round after each sampling. By contrast, in this paper, we only need a communication round per $\mathcal{O}(\epsilon^{-1})$ sampling. This is thanks to the use of batch sampling in the actor step. Also because of the batch sampling, we are able to use a constant step-size for both actor and critic steps. Here, the overall communication cost is measured by the number of communication rounds rather than the size of bits transmitted over the network. We follow the standard definition of communication complexity in the literature, which is widely adopted in the literature, see (Chen et al., 2018) (Zhang et al., 2019). However, we note that $t_{\text{gossip}}$, actor step batch size $B$ and critic iteration rounds $K_c$, scale with $\mathcal{O}(\log 1/\epsilon)$, $\mathcal{O}(1/\epsilon)$ and $\mathcal{O}(\log 1/\epsilon)$ respectively as indicated in the Theorem 3 and 2. The amount of information (in terms of bits) is $(K_c NK + BNt_{\text{gossip}})T = \mathcal{O}(\frac{N}{\epsilon^2} \log \frac{N}{\epsilon})$.

## 6 EXPERIMENTAL RESULTS

In this section, we conduct experiments to compare our proposed consensus-based TD-sharing MARL algorithm with the most related MARL algorithm 1 in Zhang et al. (2018) that also studied average reward. To our knowledge, this is the only work that is directly comparable to ours. The key difference is that the knowledge of joint action is assumed in Zhang et al. (2018), but not in our work. We vary the system size from $N = 6, 10$ to $15$. The blue curve is our TD-sharing algorithm and the red curve is classical MARL algorithm in Zhang et al. (2018). The curves represent the average results of 10 trials and the $95\%$ confidence intervals are also plotted. For the details, see A.1 in Appendix. The results in Figure 1 show that for different system sizes, both algorithms converge to a reasonable objective value. Note that if we use uniformly random policy as the baseline policy, then the objective values will be around 2 due to the setting of our experiments. All simulation results are above this threshold and our TD-sharing algorithm converge to a better objective value. See Section A.2 for addition experiment results.

## 7 CONCLUSION AND DISCUSSION

In this paper, we studied fully decentralized MARL in average reward setting and proposed a batch-sampled actor-critic algorithm. Our main contribution is to establish the first finite time convergence result for fully decentralized MARL in average reward setting, where the complexity is $\mathcal{O}(\epsilon^{-2} \log(\epsilon^{-1}))$, which matches that of the state-of-the-art single agent RL. The algorithm reaches such convergence with a better communication efficiency. However, it is still in the preliminary stage of the convergence analysis of the MARL since we only used the vanilla average. The future direction will be how to design a more scalable algorithm in terms of system size.

ACKNOWLEDGMENTS

Hairi and Jia Liu's work has been supported in part by NSF grants CAREER CNS-2110259, CNS-2112471, CNS-2102233, CCF-2110252, and a Google Faculty Research Award.

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

# A    APPENDIX

In this section, we provide lemmas that lead to the proof of both Theorem 2 and Theorem 3.

In this paper, we use $||\cdot||$ for 2-norm and $||\cdot||_{TV}$ for total variance norm. $\langle\cdot,\cdot\rangle$ denotes the inner product. Superscript $i$ in quantity $x$, i.e. $x^i$, denotes the $x$ quantity correspond to agent $i \in \mathcal{N}$. $\lambda(\cdot)$ and $\sigma(\cdot)$ denote the eigenvalues and singular values of the corresponding matrix respectively. All vectors are assumed to be column vector, unless specified. $(\cdot)^T$ is the transpose of an matrix or vector. We use $\mathbf{1}$ to denote all-1 vector with a proper dimension. For a matrix $A$, $[A]_i$ represents the $i$-th row of matrix $A$. Let $\vec{\delta} = [\delta^1, \cdots, \delta^N]^T$, i.e. the column vector for local TD-errors. In comparison, $\delta$ denotes the scalar global TD-error, i.e. $\delta = \frac{1}{N}\sum_{i\in\mathcal{N}}\delta^i$.

First, we explain the detail of our experiment setup.

## A.1    EXPERIMENT SETUP

We considered the same setting as in the Section 6.1 of Zhang et al. (2018). There are $N$ agents, each has a binary-valued action space, i.e. $\mathcal{A}^i = \{0, 1\}$, for all $i \in \mathcal{N}$. In addition, in all the results shown here, we set $|S| = 5$ states. The elements in the transition matrix are uniformly sampled from the interval $[0, 1]$ and normalized to be stochastic. We also added $10^{-5}$ onto each element to ensure ergodicity of the MDP such that the Assumption 1 is satisfied. For each agent $i$ and state action pair $(s, a)$, the mean reward $R^i(s, a)$ is sampled uniformly from $[0, 4]$. The instantaneous rewards $r^i_t$ are sampled from the uniform distribution $[R^i(s, a) - 0.5, R^i(s, a) + 0.5]$. The policy is parameterized following the Bolzmann policies, i.e.,

$$\pi^i_{\theta^i}(s, a^i) = \frac{\exp(q^T_{s,a^i}\theta^i)}{\sum_{b^i\in\mathcal{A}^i}\exp(q^T_{s,b^i}\theta^i)}$$

where $q^T_{s,b^i}$ is the feature vector with the same dimension as $\theta^i$, for all $s \in \mathcal{S}$ and $i \in \mathcal{N}$. Here, we set $m_1 = \cdots = m_N = 5$. The elements of $q_{s,b^i}$ are uniformly sampled from $[0, 1e3]$. We set the dimension for state features $K = 3$. The feature matrix $\Phi$ are insured to have full column. The stepsizes for classical MARL are set as $\beta_{w,t} = \frac{1}{t^{0.51}}$ and $\beta_{\theta,t} = \frac{1}{t^{0.52}}$. The network matrix as chosen as a ring network with diagonal elements being 0.4 and off diagonal elements 0.3 .

For our algorithm, we used step-sizes $\alpha = 1$ and $\beta = 0.1$. The batch sizes are $B = 10$, $M = 10$ and the critic iterations are 10 and actor iterations are $T = 100$.

## A.2 ADDITIONAL EXPERIMENT RESULTS

We have modified classical MARL into i)constant stepsize MARL, ii)batch MARL and iii)batch constant stepsize MARL. For classical MARL and batch MARL, we chose the stepsizes as $\beta_w = \frac{1}{t^{0.65}}$ and $\beta_\theta = \frac{1}{t^{0.85}}$ as in the paper (Zhang et al., 2018), and for the constant steptize MARL and batch constant stepsize MARL, we chose stepsizes to be $\beta_w = 0.9$ and $\beta_\theta = 0.01$. For batch MARL and batch constant stepsize MARL, we used batch size as 10. Other parameters are the same as in the A.1. We vary the system sizes from $N = 5$ to $N = 15$ and the empirical comparison results are in Figure 2 (a)-(c). In addition, we provide the comparison between average reward setting with the discounted counterpart of our algorithms for discounting factors ranging from $\gamma = 0.1$ to $\gamma = 0.999$. As we can see from Figure 2 (a)-(c), our TD-sharing algorithm performs well compared to the baseline algorithms. Among the modified algorithms, batch constant stepsize MARL shows improvement compared to the classical MARL in all three cases. Moreover in (b) and (c), when system size is larger, specifically $N = 10$ and $N = 15$, either batch modification or constant stepsize modification seem to improve the classical MARL. Yet in (a), for smaller system size, i.e. $N = 5$, only modifying to the batch size or constant stepsizes don't seem to improve the performance.
In addition, in (d), for discounted setting, as $\gamma$ increases and gets closer to 1, the objective value is closer to the average reward setting. It is because as the discounting factor approaches 1, the effective horizon, which scales with $O(\frac{\log \epsilon^{-1}}{1-\gamma})$ (Kakade, 2003), to an $\epsilon$ close stationary point gets larger and larger. As a result, it will get closer to the average reward setting. However, we can see average setting value converges to a significantly higher value. More importantly, one advantage of the average reward setting is that with more samples, the policies can potentially keep updating and so is the objective value. From Figure 2(d), we can see that as the number of sample increases, the average reward setting objective value still evolves, which means the policies are keep updating. However, for the discounted reward case, the extra sample doesn't affect the objective value.

In Figure 3, we have shown the results of different network structures on the performance when system size is 10. We compared the ring network, small world network and 2-regular network. The small world network is generated with mean node degrees being 4 and rewiring probability being 0.2. The entries of matrix $A$, for both small world network and regular network, are set as the way discussed after Assumption 3. Different network structures exhibit different performances, but all are better than baseline value 2. Among these three structures, within given sample numbers, ring network yields the best result.

## A.3 SUPPORTING LEMMAS FOR THEOREM 2

Because of the Assumption 1, by (Levin & Peres, 2017, Theorem 4.9), for aperiodic and irreducible Makrov chains, we can guarantee the following lemma holds:

**Lemma 1.** *For any policy parameter $\theta \in \mathbb{R}^{\sum_{i \in \mathcal{N}} m_i}$, consider the MDP with policy $\pi_\theta$ and transition kernel $P(\cdot|s, a)$. Let $d_\theta$ be the stationary distribution of the MDP. There exist constants $\kappa > 0$ and $\rho \in (0, 1)$ such that $\sup_{s \in \mathcal{S}} ||P(s_t|s_0 = s) - d_\theta||_{TV} \leq \kappa \rho^t, \quad \forall t \geq 0$.*

This lemma has been adopted directly as an assumption in many related works in theoretical analysis of RL (Xu et al., 2020; Chen et al., 2021; Qiu et al., 2021).

As a result of the Lemma 1, the Markov chain of state-action pair $\{s_t, a_t\}_{t \geq 0}$ for policy $\pi_\theta$ also has the property of ergodicity. We state this result as the following lemma.

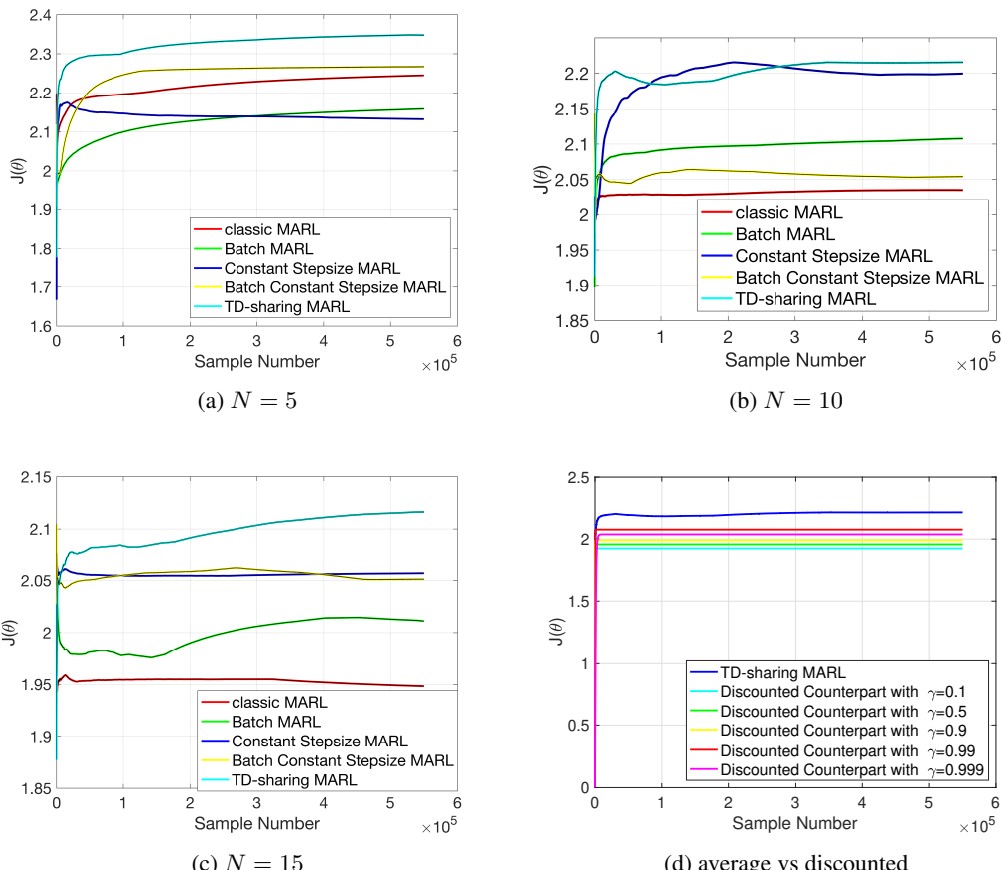

(a) $N = 5$

(b) $N = 10$

(c) $N = 15$

(d) average vs discounted

Figure 2: The empirical comparisons of algorithms

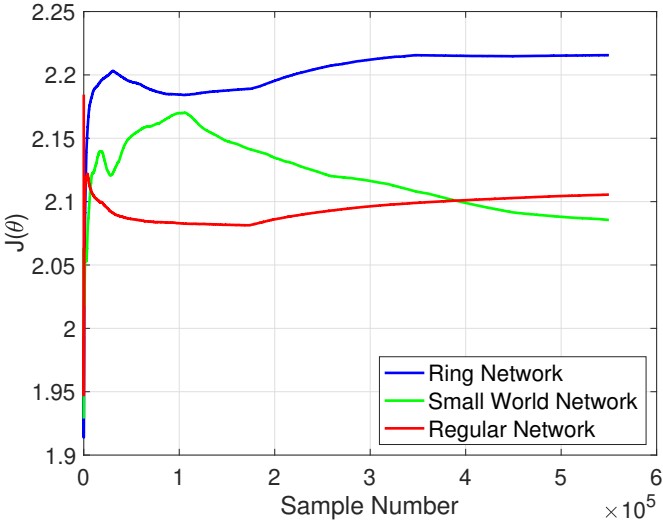

Figure 3: Different Network Structures on Performance When $N = 10$

**Lemma 2.** *Suppose the Assumption 1 hold, let $\nu_\theta$ be the stationary distribution of the state-action pair MDP. Then, we have*

$$\sup_{s \in \mathcal{S}} ||P(s_t, a_t|s_0 = s) - \nu_\theta||_{TV} \leq \kappa \rho^t, \quad \forall t \geq 0. \tag{10}$$

*Proof.* For any given $s_0 \in \mathcal{S}$, by definition, we have

$$
\begin{aligned}
||P(s_t, a_t|s_0) - \nu_\theta||_{TV} &= \frac{1}{2} \sum_{(s,a) \in \mathcal{S} \times \mathcal{A}} |P(s_t = s, a_t = a|s_0) - \nu_\theta(s,a)| \\
&= \frac{1}{2} \sum_{(s,a) \in \mathcal{S} \times \mathcal{A}} |P(s_t = s|s_0)\pi_\theta(a_t = a|s_t = s) - \nu_\theta^s(s)\pi_\theta(a|s)| \\
&= \frac{1}{2} \sum_{(s,a) \in \mathcal{S} \times \mathcal{A}} |(P(s_t = s|s_0) - \nu_\theta^s(s))\pi_\theta(a|s)| \\
&= \frac{1}{2} \sum_{(s,a) \in \mathcal{S} \times \mathcal{A}} \pi_\theta(a|s)|P(s_t = s|s_0) - \nu_\theta^s(s)| \\
&= \frac{1}{2} \sum_{s \in \mathcal{S}} |P(s_t = s|s_0) - \nu_\theta^s(s)| \\
&= ||P(s_t|s_0) - \nu_\theta^s||_{TV} \leq \kappa \rho^t.
\end{aligned}
$$

Since it holds for all $s_0 \in \mathcal{S}$, equation 10 holds. $\square$

As a result of Assumption 5, we have the following lemmas.

**Lemma 3.** *For all policy $\pi_\theta$, we have*

$$\langle w, A_{\pi_\theta} w \rangle \leq -\frac{\lambda_A}{2} ||w||^2 \tag{11}$$

*where $\lambda_A$ is defined in Assumption 5.*

*Proof.* Because of the fact $w^T A_{\pi_\theta} w = w^T A_{\pi_\theta}^T w$, we have

$$
\begin{aligned}
w^T A_{\pi_\theta} w &= \frac{1}{2}(w^T A_{\pi_\theta} w + w^T A_{\pi_\theta}^T w) \\
&= \frac{1}{2} w^T (A_{\pi_\theta} + A_{\pi_\theta}^T) w \\
&\leq -\frac{\lambda_A}{2} w^T w = -\frac{\lambda_A}{2} ||w||^2. \tag{12}
\end{aligned}
$$

$\square$

In fact, interestingly enough, the Assumption 5 and Lemma 3 are equivalent in a sense that if the statement in the Lemma 3 is taken as the assumption, the statement in the Assumption 5 can be obtained as a result. And the paper (Xu et al., 2020) used the statement in the Lemma 3 as an assumption, whereas in ours and (Qiu et al., 2021), we assumed Assumption 5.

**Lemma 4.** *For any given policy $\pi_\theta$, the corresponding optimal value function approximation parameter $w_\theta^*$ is uniformly bounded, specifically, there exists $R_w := \frac{4r_{\max}}{\lambda_A} > 0$ such that*

$$||w_\theta^*|| \leq R_w. \tag{13}$$

*Proof.* It's easy to see that $J(\theta) \leq r_{\max}$ from equation 2 and $\bar{r}(s,a) \leq r_{\max}$ for any $(s,a)$ pair. Then, we have

$$||w_\theta^*|| = || - A_{\pi_\theta}^{-1} b_{\pi_\theta}||$$

$$= || - \left(\mathbb{E}_{s \sim d_\theta(s), s' \sim P(\cdot|s)}[(\phi(s') - \phi(s))\phi^T(s)]\right)^{-1} \cdot \mathbb{E}_{s \sim d_\theta, a \sim \pi_\theta}[\phi(s)(\bar{r}(s,a) - J(\theta))]||$$

$$\leq || - \left(\mathbb{E}_{s \sim d_\theta(s), s' \sim P(\cdot|s)}[(\phi(s') - \phi(s))\phi^T(s)]\right)^{-1} || \cdot ||\mathbb{E}_{s \sim d_\theta, a \sim \pi_\theta}[\phi(s)(\bar{r}(s,a) - J(\theta))]||$$

$$= \frac{1}{\sigma_{\min}\left(-\mathbb{E}_{s \sim d_\theta(s), s' \sim P(\cdot|s)}[(\phi(s') - \phi(s))\phi^T(s)]\right)} \cdot ||\mathbb{E}_{s \sim d_\theta, a \sim \pi_\theta}[\phi(s)(\bar{r}(s,a) - J(\theta))]||$$

$$\leq \frac{2||\mathbb{E}_{s \sim d_\theta, a \sim \pi_\theta}[\phi(s)(\bar{r}(s,a) - J(\theta))]||}{\lambda_{\min}(-A_{\pi_\theta} - A_{\pi_\theta}^T)}$$

$$\leq \frac{2\mathbb{E}_{s \sim d_\theta, a \sim \pi_\theta}[||\phi(s)|| \cdot (|\bar{r}(s,a)| + |J(\theta)|)]}{\lambda} = \frac{4r_{\max}}{\lambda_A}$$

where the third equality used the fact $||A^{-1}|| = \frac{1}{\sigma_{\min}(A)}$ and the second from the last inequality is from Bhatia (1997) (Proposition III 5.1) . $\qquad\square$

Note that, for agent $i \in \mathcal{N}$, the estimated long term average reward $\mu_{k,\tau}^i$ at sample $\tau$ of iteration $k$ in equation 5 can be written as

$$\mu_{k,\tau}^i = \beta \sum_{l=1}^{\tau}(1-\beta)^{\tau-l}r_{k,l}^i + (1-\beta)^\tau \mu_0^i. \tag{14}$$

**Lemma 5.** *For any $i \in \mathcal{N}$ and $t \geq 0$, step size $0 < \beta < 1$, for the estimated long term average reward for agent $i$, we have*

$$\mu_{t+1}^i = (1-\beta)\mu_t^i + \beta r_t^i \tag{15}$$

*is bounded by $r_{\max}$, i.e. $|\mu_{t+1}^i| \leq r_{\max}$.*

*Proof.* WLOG, we suppose that $0 < \mu_0^i \leq r_{\max}$, we have

$$|\mu_{t+1}^i| = |(1-\beta)\mu_t^i + \beta r_t^i|$$
$$\leq (1-\beta)|\mu_t^i| + \beta|r_t^i|. \tag{16}$$

By the supposition, we have $|\mu_0^i| \leq r_{\max}$. We assume $|\mu_t^i| \leq r_{\max}$ holds for iteration $t > 0$, then for $t+1$, by equation 16

$$|\mu_{t+1}^i| \leq (1-\beta)|\mu_t^i| + \beta|r_t^i| \leq (1-\beta)r_{\max} + \beta r_{\max} = r_{\max}.$$

Therefore, Lemma 5 holds by mathematical induction. $\qquad\square$

For a given policy $\pi_\theta$, to establish a bound on the difference between the optimal approximation parameter $w_\theta^*$, we first derive a bound the difference between parameter $w_k^i$ and the average among all agents $\bar{w}_k$ at time $k$. Then, we derive a bound for the difference between average $\bar{w}_k$ and the optimal $w_\theta^*$.

We have following notations for the analysis. Given an agent $i \in \mathcal{N}$, we consider the consensus error at time $k$ and we denote $Q_k^i = w_k^i - \bar{w}_k$, where $\bar{w}_k := \frac{1}{N}\sum_{i \in \mathcal{N}} w_k^i$. Then, we denote the matrix form as $\mathbf{Q}_k = [Q_k^1, \cdots, Q_k^N] \in \mathbb{R}^{K \times N}$ Then, we have the following lemma.

**Lemma 6.** *Suppose the Assumption 2 and 3 hold. For the consensus error matrix, we have*

$$||\mathbf{Q}_{k+1}|| \leq \kappa_1 N^2 \gamma^k ||\mathbf{Q}_0|| + \kappa_2' N^3 \beta \tag{17}$$

*where $\gamma := (1 - \eta^{N-1}) \cdot (1 + 2\beta) < 1$, $\kappa_1 = 2(1+2\beta)(1+\eta^{-(N-1)})$ and $\kappa_2' = \frac{4(1+\eta^{-(N-1)})r_{\max}}{1-\gamma}$.*

*Proof.* By equation 7, the parameter update $\tilde{w}_k^i$ for agent $i$ at iteration $k$ can be written as follows

$$\tilde{w}_k^i = w_k^i + \frac{\beta}{M}\sum_{\tau=0}^{M}\delta_{k,\tau}^i \cdot \phi(s_{k,\tau})$$

$$= w_k^i + \frac{\beta}{M}\sum_{\tau=0}^{M-1}[r_{k,\tau+1}^i - \beta\sum_{l=1}^{\tau}(1-\beta)^{\tau-l}r_{k,l}^i - (1-\beta)^\tau\mu_0^i$$

$$+ \phi(s_{k,\tau+1})^T w_k^i - \phi(s_{k,\tau})^T w_k^i] \cdot \phi(s_{k,\tau})$$

$$= w_k^i + \frac{\beta}{M} \sum_{\tau=0}^{M-1} \phi(s_{k,\tau})[\phi(s_{k,\tau+1}) - \phi(s_{k,\tau})]^T w_k^i$$

$$+ \frac{\beta}{M} \sum_{\tau=0}^{M-1} (r_{k,\tau+1}^i - \beta \sum_{l=1}^{\tau} (1-\beta)^{\tau-l} r_{k,l}^i - (1-\beta)^{\tau} \mu_0^i) \phi(s_{k,\tau}). \tag{18}$$

After a consensus step, the update will be

$$w_{k+1}^i = \sum_{j \in \mathcal{N}_i} A(i,j) \cdot \tilde{w}_k^j$$

$$= \sum_{j \in \mathcal{N}_i} A(i,j) \cdot [w_k^j + \frac{\beta}{M} \sum_{\tau=0}^{M-1} \phi(s_{k,\tau})[\phi(s_{k,\tau+1}) - \phi(s_{k,\tau})]^T w_k^j$$

$$+ \frac{\beta}{M} \sum_{\tau=0}^{M-1} (r_{k,\tau+1}^j - \beta \sum_{l=1}^{\tau} (1-\beta)^{\tau-l} r_{k,l}^j - (1-\beta)^{\tau} \mu_0^j) \phi(s_{k,\tau})]$$

$$= \sum_{j \in \mathcal{N}_i} A(i,j) \cdot w_k^j + \frac{\beta}{M} \sum_{\tau=0}^{M-1} \phi(s_{k,\tau})[\phi(s_{k,\tau+1}) - \phi(s_{k,\tau})]^T \sum_{j \in \mathcal{N}_i} A(i,j) \cdot w_k^j$$

$$+ \frac{\beta}{M} \sum_{\tau=0}^{M-1} \phi(s_{k,\tau}) \sum_{j \in \mathcal{N}_i} A(i,j) \cdot (r_{k,\tau+1}^j - \beta \sum_{l=1}^{\tau} (1-\beta)^{\tau-l} r_{k,l}^j - (1-\beta)^{\tau} \mu_0^j)$$

$$= \mathbf{w}_k \cdot (A_{i,:})^T + \frac{\beta}{M} \sum_{\tau=0}^{M-1} \phi(s_{k,\tau})[\phi(s_{k,\tau+1}) - \phi(s_{k,\tau})]^T \mathbf{w}_k \cdot (A_{i,:})^T$$

$$+ \frac{\beta}{M} \sum_{\tau=0}^{M-1} \phi(s_{k,\tau}) A_{i,:} (r_{k,\tau+1} - \beta \sum_{l=1}^{\tau} (1-\beta)^{\tau-l} r_{k,l} - (1-\beta)^{\tau} \mu_0) \tag{19}$$

where $\mathbf{w}_k = [w_k^1, w_k^2, \cdots, w_k^N] \in \mathbb{R}^{K \times N}$ is the matrix form of all parameters at time $k$, $A_{i,:}$ is the $i$-th row of matrix $A$ and $r_{k,l} = (r_{k,l}^1, \cdots, r_{k,l}^N)$. Now we consider the average dynamics of the algorithm. Recall $\bar{w}_k = \frac{1}{N} \sum_{i \in \mathcal{N}} w_k^i$, then using equation 19 we have

$$\bar{w}_{k+1} = \frac{1}{N} \sum_{i \in \mathcal{N}} w_{k+1}^i$$

$$= \frac{1}{N} \sum_{i \in \mathcal{N}} \left( \mathbf{w}_k \cdot (A_{i,:})^T + \frac{\beta}{M} \sum_{\tau=0}^{M-1} \phi(s_{k,\tau})[\phi(s_{k,\tau+1}) - \phi(s_{k,\tau})]^T \mathbf{w}_k \cdot (A_{i,:})^T \right.$$

$$\left. + \frac{\beta}{M} \sum_{\tau=0}^{M-1} \phi(s_{k,\tau}) A_{i,:} (r_{k,\tau+1} - \beta \sum_{l=1}^{\tau} (1-\beta)^{\tau-l} r_{k,l} - (1-\beta)^{\tau} \mu_0) \right)$$

$$= \frac{1}{N} \mathbf{w}_k \cdot \mathbf{1} + \frac{\beta}{M} \sum_{\tau=0}^{M-1} \phi(s_{k,\tau})[\phi(s_{k,\tau+1}) - \phi(s_{k,\tau})]^T \frac{1}{N} \mathbf{w}_k \cdot \mathbf{1}$$

$$+ \frac{\beta}{M} \sum_{\tau=0}^{M-1} \phi(s_{k,\tau}) \frac{1}{N} \sum_{i \in \mathcal{N}} A_{i,:} (r_{k,\tau+1} - \beta \sum_{l=1}^{\tau} (1-\beta)^{\tau-l} r_{k,l} - (1-\beta)^{\tau} \mu_0)$$

$$= \bar{w}_k + \frac{\beta}{M} \sum_{\tau=0}^{M-1} \phi(s_{k,\tau})[\phi(s_{k,\tau+1}) - \phi(s_{k,\tau})]^T \bar{w}_k$$

$$+ \frac{\beta}{M} \sum_{\tau=0}^{M-1} \phi(s_{k,\tau}) \frac{1}{N} \mathbf{1}^T (r_{k,\tau+1} - \beta \sum_{l=1}^{\tau} (1-\beta)^{\tau-l} r_{k,l} - (1-\beta)^{\tau} \mu_0)$$

$$= \bar{w}_k + \frac{\beta}{M} \sum_{\tau=0}^{M-1} \phi(s_{k,\tau})[\phi(s_{k,\tau+1}) - \phi(s_{k,\tau})]^T \bar{w}_k$$

$$+ \frac{\beta}{M} \sum_{\tau=0}^{M-1} \phi(s_{k,\tau})(\bar{r}_{k,\tau+1} - \beta \sum_{l=1}^{\tau} (1-\beta)^{\tau-l} \bar{r}_{k,l} - (1-\beta)^\tau \mu_0) \tag{20}$$

where $\bar{r}_{k,l} = \frac{1}{N} \sum_{i \in \mathcal{N}} r_{k,l}^i$.

Given an agent $i \in \mathcal{N}$, we consider the consensus error at time $k$ and recall $Q_k^i = w_k^i - \bar{w}_k$. Then, we have

$$Q_{k+1}^i = w_{k+1}^i - \bar{w}_{k+1}$$

$$= \mathbf{w}_k \cdot (A_{i,:})^T - \bar{w}_k + \frac{\beta}{M} \sum_{\tau=0}^{M-1} \phi(s_{k,\tau})[\phi(s_{k,\tau+1}) - \phi(s_{k,\tau})]^T (\mathbf{w}_k \cdot (A_{i,:})^T - \bar{w}_k)$$

$$+ \frac{\beta}{M} \sum_{\tau=0}^{M-1} \phi(s_{k,\tau})(A_{i,:} - \frac{1}{N} \mathbf{1}^T)(r_{k,\tau+1} - \beta \sum_{l=1}^{\tau} (1-\beta)^{\tau-l} r_{k,l} - (1-\beta)^\tau \mu_0). \tag{21}$$

Then, for the matrix form $\mathbf{Q}_k = [Q_k^1, \cdots, Q_k^N] \in \mathbb{R}^{K \times N}$, we have

$$\mathbf{Q}_{k+1} = \mathbf{Q}_k A^T + \frac{\beta}{M} \sum_{\tau=0}^{M-1} \phi(s_{k,\tau})[\phi(s_{k,\tau+1}) - \phi(s_{k,\tau})]^T \mathbf{Q}_k A^T$$

$$+ \frac{\beta}{M} \sum_{\tau=0}^{M-1} \phi(s_{k,\tau})[(A - \frac{1}{N} \mathbf{1}\mathbf{1}^T)(r_{k,\tau+1} - \beta \sum_{l=1}^{\tau} (1-\beta)^{\tau-l} r_{k,l} - (1-\beta)^\tau \mu_0)]^T$$

$$= \mathbf{Q}_k A^T + \frac{\beta}{M} \sum_{\tau=0}^{M-1} \phi(s_{k,\tau})[\phi(s_{k,\tau+1}) - \phi(s_{k,\tau})]^T \mathbf{Q}_k A^T$$

$$+ \frac{\beta}{M} \sum_{\tau=0}^{M-1} \phi(s_{k,\tau})[r_{k,\tau+1} - \beta \sum_{l=1}^{\tau} (1-\beta)^{\tau-l} r_{k,l} - (1-\beta)^\tau \mu_0]^T (A - \frac{1}{N} \mathbf{1}\mathbf{1}^T)$$

$$\tag{22}$$

where the first equality is due to $A$ being doubly stochastic. For convenience, denote $B_k = \frac{1}{M} \sum_{\tau=0}^{M-1} \phi(s_{k,\tau})[\phi(s_{k,\tau+1}) - \phi(s_{k,\tau})]^T$ and $C_k = \frac{1}{M} \sum_{\tau=0}^{M-1} \phi(s_{k,\tau})[r_{k,\tau+1} - \beta \sum_{l=1}^{\tau} (1-\beta)^{\tau-l} r_{k,l} - (1-\beta)^\tau \mu_0]^T$. Then, iteratively we have

$$\mathbf{Q}_{k+1} = (I + \beta B_k) \mathbf{Q}_k A^T + \beta C_k (A - \frac{1}{N} \mathbf{1}\mathbf{1}^T)$$

$$= \prod_{t=0}^{k} (I + \beta B_t) \mathbf{Q}_0 (A^T)^{k+1} + \beta \sum_{t=0}^{k} \prod_{\tilde{t}>t}^{k} (I + \beta B_{\tilde{t}}) C_t (A - \frac{1}{N} \mathbf{1}\mathbf{1}^T) A^{k-t} \tag{23}$$

Then, the norm of the consensus error is following

$$||\mathbf{Q}_{k+1}|| = || \prod_{t=0}^{k} (I + \beta B_t) \mathbf{Q}_0 (A^T)^{k+1} + \beta \sum_{t=0}^{k} \prod_{\tilde{t}>t}^{k} (I + \beta B_{\tilde{t}}) C_t (A - \frac{1}{N} \mathbf{1}\mathbf{1}^T) A^{k-t}||$$

$$\leq || \prod_{t=0}^{k} (I + \beta B_t)|| \cdot ||\mathbf{Q}_0 (A^T)^{k+1}|| + \beta \sum_{t=0}^{k} \prod_{\tilde{t}>t}^{k} ||(I + \beta B_{\tilde{t}})|| \cdot ||C_t|| \cdot ||(A - \frac{1}{N} \mathbf{1}\mathbf{1}^T) A^{k-t}||.$$

$$\tag{24}$$

Note that since $A$ is doubly stochastic, so is $A^T$. Let $(A^T)_{:,i}^{k+1}$ be the $i$-th column of matrix $(A^T)^{k+1}$. From Nedic & Ozdaglar (2009), we know that

$$||\mathbf{Q}_0 (A^T)_{:,i}^{k+1}|| = ||\mathbf{Q}_0 (A^T)_{:,i}^{k+1} - \mathbf{Q}_0 \frac{1}{N} \mathbf{1}||$$

$$= || \sum_{j \in \mathcal{N}} ((A^T)^{k+1}_{j,i} - \frac{1}{N}) Q^j_0 ||$$

$$\leq \sum_{j \in \mathcal{N}} |(A^T)^{k+1}_{j,i} - \frac{1}{N}| \cdot ||Q^j_0||$$

$$\leq N \cdot 2 \frac{1 + \eta^{-(N-1)}}{1 - \eta^{N-1}} (1 - \eta^{N-1})^{k+1} \cdot \max_{j \in \mathcal{N}} ||Q^j_0||$$

$$\leq 2N(1 + \eta^{-(N-1)})(1 - \eta^{N-1})^k \cdot ||\mathbf{Q}_0||. \tag{25}$$

Hence, $||\mathbf{Q}_0 A^{k+1}|| \leq 2N^2(1 + \eta^{-(N-1)})(1 - \eta^{N-1})^k \cdot ||\mathbf{Q}_0||$. To bound the first term in the RHS of equation 24, we have

$$|| \prod_{t=0}^{k} (I + \beta B_t)|| \leq \prod_{t=0}^{k} ||I + \beta B_t||$$

$$\leq \prod_{t=0}^{k} (||I|| + \beta ||B_t||)$$

$$\leq \prod_{t=0}^{k} (1 + 2\beta) = (1 + 2\beta)^{k+1}. \tag{26}$$

In order to make sure it converges to the neighborhood of a consensus in the limit, the step size has to satisfy $\gamma := (1 - \eta^{N-1}) \cdot (1 + 2\beta) < 1$, which results in $\beta < \frac{\eta^{N-1}}{2(1 - \eta^{N-1})}$. Hence, for the first term in the RHS of equation 24, we have

$$|| \prod_{t=0}^{k} (I + \beta B_t)|| \cdot ||\mathbf{Q}_0 A^{k+1}|| \leq \kappa_1 N^2 \gamma^k ||\mathbf{Q}_0||$$

where $\kappa_1 = 2(1 + 2\beta)(1 + \eta^{-(N-1)})$.

Furthermore, we have

$$||(A - \frac{1}{N} \mathbf{1}\mathbf{1}^T) A^{k-t}|| = ||A^{k-t+1} - \frac{1}{N} \mathbf{1}\mathbf{1}^T|| \leq 2N^2(1 + \eta^{-(N-1)})(1 - \eta^{N-1})^{k-t}. \tag{27}$$

And $||C_t|| \leq 2N r_{\max}$, where $r_{\max} = \sup_{i,s,a} r^i(s,a)$. This is because

$$||C_k|| = ||\frac{1}{M} \sum_{\tau=0}^{M-1} \phi(s_{k,\tau})[r_{k,\tau+1} - \beta \sum_{l=1}^{\tau} (1 - \beta)^{\tau-l} r_{k,l} - (1 - \beta)^\tau \mu_0]^T ||$$

$$\leq \frac{1}{M} \sum_{\tau=0}^{M-1} ||\phi(s_{k,\tau})[r_{k,\tau+1} - \beta \sum_{l=1}^{\tau} (1 - \beta)^{\tau-l} r_{k,l} - (1 - \beta)^\tau \mu_0]^T ||$$

$$\leq \frac{1}{M} \sum_{\tau=0}^{M-1} ||\phi(s_{k,\tau})|| \cdot ||r_{k,\tau+1} - \beta \sum_{l=1}^{\tau} (1 - \beta)^{\tau-l} r_{k,l} - (1 - \beta)^\tau \mu_0||$$

$$\leq \frac{1}{M} \sum_{\tau=0}^{M-1} [||r_{k,\tau+1}|| + \beta \sum_{l=1}^{\tau} (1 - \beta)^{\tau-l} ||r_{k,l}|| + (1 - \beta)^\tau ||\mu_0||]$$

$$\leq \frac{1}{M} \sum_{\tau=0}^{M-1} [N r_{\max} + \beta \frac{1 - (1 - \beta)^\tau}{1 - (1 - \beta)} N r_{\max} + (1 - \beta)^\tau N r_{\max}] = 2N r_{\max}.$$

Then for the second term of equation 24, we have

$$\beta \sum_{t=0}^{k} \prod_{\tilde{t} > t}^{k} ||(I + \beta B_{\tilde{t}})|| \cdot ||C_t|| \cdot ||(A - \frac{1}{N} \mathbf{1}\mathbf{1}^T) A^{k-t}||$$

$$\leq \beta \sum_{t=0}^{k} (1+2\beta)^{k-t} \cdot 2Nr_{\max} \cdot 2N^2 (1+\eta^{-(N-1)})(1-\eta^{N-1})^{k-t}$$

$$= \kappa_2 \beta \sum_{t=0}^{k} \gamma^{k-t} = \kappa_2 N^3 \beta \frac{1-\gamma^{k+1}}{1-\gamma} \tag{28}$$

where $\kappa_2 = 4(1+\eta^{-(N-1)})r_{\max}$. Further, for consensus error equation 24, we have

$$||\mathbf{Q}_{k+1}|| \leq \kappa_1 N^2 \gamma^k ||\mathbf{Q}_0|| + \kappa_2 N^3 \beta \frac{1-\gamma^{k+1}}{1-\gamma} \leq \kappa_1 N^2 \gamma^k ||\mathbf{Q}_0|| + \kappa_2' N^3 \beta \tag{29}$$

where $\kappa_2' = \frac{\kappa_2}{1-\gamma}$. $\qquad\square$

The corresponding average parameter under the policy $\pi_\theta$ will converge to the solution to the following equation $w_\theta^* = A_\theta^{-1} b_\theta$, where $A_\theta = \mathbb{E}_{s \sim d_\theta(s)}[\phi(s)(\phi(s') - \phi(s))^T]$, $b_\theta = \mathbb{E}_{(s,a) \sim d_\theta(s,a)}[(\bar{r}(s,a) - J(\theta))\phi(s)]$ and $\bar{r}(s,a) = \frac{1}{N} \sum_{i \in \mathcal{N}} r^i(s,a)$.

Now consider $||\bar{w}_k - w^*||$. Recall the from the average parameter equation equation 20 and the corresponding ODE is

$$\dot{w}_\theta = A_\theta w_\theta + b_\theta. \tag{30}$$

For the difference between average dynamics and the optimal value, we have the following lemma.

**Lemma 7.** *For* $\beta \leq \min\{\frac{\lambda_A}{128}, \frac{4}{\lambda_A}\}$ *and* $M \geq (\frac{1}{\lambda_A} + \beta)\frac{6144[1+(\kappa-1)\rho]}{(1-\rho)\lambda_A}$, *we have*

$$\mathbb{E}[||\bar{w}_K - w_\theta^*||_2^2] \leq (1 - \frac{\lambda_A}{8}\beta)^K ||\bar{w}_0 - w_\theta^*||_2^2 + (\frac{1}{\lambda_A} + \beta)\frac{1536(4R_w^2 + r_{\max}^2)[1+(\kappa-1)\rho]}{(1-\rho)\lambda_A M}. \tag{31}$$

*Proof.* The proof follows from verifying the Assumption 3 of Xu et al. (2020), then we can apply the results from Theorem 4 of Xu et al. (2020).

1. For item 1 in the assumption, it's easy to check that $||A_\theta|| \leq C_A = 4$ by Assumption 4 and $b_\theta \leq C_b = 2r_{\max}$.

2. For item 2, it holds because of Lemma 3.

3. For item 3, it holds because of Lemma 1.

And we recall that the bound on $||w_\theta^*||$ is $R_w = \frac{4r_{\max}}{\lambda_A}$. Hence, by Theorem 4 of Xu et al. (2020), $\beta \leq \min\{\frac{\lambda_A}{8C_A^2}, \frac{4}{\lambda_A}\}$ and $M \geq (\frac{2}{\lambda_A} + 2\beta)\frac{192C_A^2[1+(\kappa-1)\rho]}{(1-\rho)\lambda_A}$, equation 31 holds. $\qquad\square$

## A.4 Proof of Theorem 2

As a result from Lemma 4 and Lemma 7, we provide the following proof for Theorem 2.

*Proof.* Therefore, by equation 17 and equation 31, we have

$$\mathbb{E}[\sum_{i=1}^{N} ||w_K^i - w_\theta^*||^2]$$

$$\leq 2\mathbb{E}[\sum_{i=1}^{N} ||w_K^i - \bar{w}_K||^2] + 2\mathbb{E}[\sum_{i=1}^{N} ||\bar{w}_K - w_\theta^*||^2]$$

$$\leq 2||\mathbf{Q}_K||^2 + 2N\mathbb{E}[||\bar{w}_K - w_\theta^*||^2]$$

$$\leq 2(\kappa_1 N^2 \gamma^K ||\mathbf{Q}_0|| + \kappa_2' N^3 \beta)^2 + 2N(1 - \frac{\lambda_A}{8}\beta)^K ||\bar{w}_0 - w_\theta^*||_2^2$$

$$+ 2N(\frac{2}{\lambda_A} + 2\beta)\frac{192(C_A^2 R_w^2 + C_b^2)[1 + (\kappa - 1)\rho]}{(1 - \rho)\lambda_A M}$$

$$\leq \kappa_1' N^4 \gamma^{2K} + \kappa_2'' N^6 \beta^2 + \kappa_3 N^5 \gamma^K \beta + 2N(1 - \frac{\lambda_A}{8}\beta)^K ||\bar{w}_0 - w_\theta^*||_2^2 + \frac{\kappa_4}{M}N$$

where $\kappa_1' = 2\kappa_1^2 ||\mathbf{Q}_0||^2$, $\kappa''_2 = 2\kappa_2'^2$, $\kappa_3 = 4\kappa_1\kappa_2'||\mathbf{Q}_0||$ and $\kappa_4 = (\frac{1}{\lambda_A} + \beta)\frac{3072(4R_w^2 + r_{\max}^2)[1 + (\kappa - 1)\rho]}{(1 - \rho)\lambda_A}$. $\qquad\square$

## A.5 PROOF OF THEOREM 3

We use $w_t^*$ to denote the optimal value function parameter under policy $\theta_t$ at time $t$ and $w_t = [(w_t^1)^T, \cdots, (w_t^N)^T]^T \in \mathbb{R}^{N \times K}$, which is the aggregated function approximation parameters from Line 2 of Algorithm 2. Recall the $\tilde{\delta}_{t,l}^i$ generated by Line 18 of Algorithm 2, which is the $i$-th agent's estimate of the global TD-error from sample $l$ at time $t$ after consensus. Let $v_t^i(w_t) = \frac{1}{B}\sum_{l=0}^{B-1} \tilde{\delta}_{t,l}^i(w_t) \cdot \psi_{t,l}^i$ and $v_t(w_t) = [(v_t^1(w_t))^T, \cdots, (v_t^N(w_t))^T]^T$, $h_t^i(w_t^*) = \frac{1}{B}\sum_{l=0}^{B-1} \delta_{t,l}(w_t^*) \cdot \psi_{t,l}^i$ and $h_t(w_t^*) = [(h_t^1(w_t^*))^T, \cdots, (h_t^N(w_t^*))^T]^T$, $\text{Adv}_w(s,a) = \mathbb{E}_{s' \sim P(\cdot|s,a), r \sim d_r(s,a)}[\delta_w(s,a,s')|s,a]$ and $g^i(w,\theta) = \mathbb{E}[\text{Adv}_w(s,a)\psi_{\theta^i}(s,a^i)]$, where $d_r(s,a)$ is the reward distribution of state-action pair $(s,a)$. By Taylor expansion and the Lipschitz property from Assumption 6, we have

$$J(\theta_{t+1}) \geq J(\theta_t) + \langle \nabla_\theta J(\theta_t), \theta_{t+1} - \theta_t \rangle - \frac{L_J}{2}||\theta_{t+1} - \theta_t||^2$$

$$= J(\theta_t) + \alpha \langle \nabla_\theta J(\theta_t), v_t(w_t) - \nabla_\theta J(\theta_t) + \nabla_\theta J(\theta_t) \rangle - \frac{L_J \alpha^2}{2}||v_t(w_t)||^2$$

$$= J(\theta_t) + \alpha ||\nabla_\theta J(\theta_t)||^2 + \alpha \langle \nabla_\theta J(\theta_t), v_t(w_t) - \nabla_\theta J(\theta_t) \rangle - \frac{L_J \alpha^2}{2}||v_t(w_t)$$
$$- \nabla_\theta J(\theta_t) + \nabla_\theta J(\theta_t)||^2$$

$$\geq J(\theta_t) + (\frac{1}{2}\alpha - L_J\alpha^2)||\nabla_\theta J(\theta_t)||^2 - (\frac{1}{2}\alpha + L_J\alpha^2)||v_t(w_t) - \nabla_\theta J(\theta_t)||^2$$

where the last inequality is because

$$\langle \nabla_\theta J(\theta_t), v_t(w_t) - \nabla_\theta J(\theta_t) \rangle \geq -\frac{1}{2}||\nabla_\theta J(\theta_t)||^2 - \frac{1}{2}||v_t(w_t) - \nabla_\theta J(\theta)||^2,$$

and

$$||v_t(w_t) - \nabla_\theta J(\theta_t) + \nabla_\theta J(\theta_t)||^2 \leq 2||v_t(w_t) - \nabla_\theta J(\theta_t)||^2 + 2||\nabla_\theta J(\theta_t)||^2. \qquad (32)$$

Taking expectations on both sides conditioned on the filtration $\mathcal{F}_t$ and rearranging the terms, we have

$$(\frac{1}{2}\alpha - L_J\alpha^2)\mathbb{E}[||\nabla_\theta J(\theta_t)||^2|\mathcal{F}_t] \leq \mathbb{E}[J(\theta_{t+1})|\mathcal{F}_t] - J(\theta_t)$$
$$+ (\frac{1}{2}\alpha + L_J\alpha^2)\mathbb{E}[||v_t(w_t) - \nabla_\theta J(\theta_t)||^2|\mathcal{F}_t]. \qquad (33)$$

Then, we establish upper bound on the third term of the RHS. By definition, we have

$$||v_t(w_t) - \nabla_\theta J(\theta_t)||^2$$
$$= ||v_t(w_t) - v_t(w_t^*) + v_t(w_t^*) - h_t(w_t^*) + h_t(w_t^*) - g(w_t^*, \theta_t) + g(w_t^*, \theta_t) - \nabla_\theta J(\theta_t)||^2$$
$$\leq 6||v_t(w_t) - v_t(w_t^*)||^2 + 6||v_t(w_t^*) - h_t(w_t^*)||^2 + 6||h_t(w_t^*) - g(w_t^*, \theta_t)||^2$$
$$+ 6||g(w_t^*, \theta_t) - \nabla_\theta J(\theta_t)||^2. \qquad (34)$$

We note that $||v_t(w_t) - v_t(w_t^*)||^2 = \sum_{i \in \mathcal{N}} ||v_t^i(w_t) - v_t^i(w_t^*)||^2$ and the other three terms in equation 34 can also be similarly decomposed. For the first term in the RHS of equation 34, we have

$$||v_t^i(w_t) - v_t^i(w_t^*)||^2$$

$$= ||\frac{1}{B}\sum_{l=0}^{B-1}\tilde{\delta}_{t,l}^{i}(w_t)\cdot\psi_{t,l}^{i} - \frac{1}{B}\sum_{l=0}^{B-1}\tilde{\delta}_{t,l}^{i}(w_t^*)\cdot\psi_{t,l}^{i}||^2$$

$$= ||\frac{1}{B}\sum_{l=0}^{B-1}[\tilde{\delta}_{t,l}^{i}(w_t) - \tilde{\delta}_{t,l}^{i}(w_t^*)]\cdot\psi_{t,l}^{i}||^2$$

$$= ||\frac{1}{B}\sum_{l=0}^{B-1}[A^{t_\text{gossip}}]_i(\mathbf{w}_t - w_t^*\mathbf{1}^T)^T[\phi(s_{t,l+1}) - \phi(s_{t,l})]\cdot\psi_{t,l}^{i}||^2$$

$$\leq \max_{l\in\{0,\cdots,B-1\}}||[A^{t_\text{gossip}}]_i(\mathbf{w}_t - w_t^*\mathbf{1}^T)^T[\phi(s_{t,l+1}) - \phi(s_{t,l})]\cdot\psi_{t,l}^{i}||^2$$

$$\leq \max_{l\in\{0,\cdots,B-1\}}||[A^{t_\text{gossip}}]_i||^2||[\phi(s_{t,l+1}) - \phi(s_{t,l})]||^2||\mathbf{w}_t - w_t^*\mathbf{1}^T||^2\cdot||\psi_{t,l}^{i}||^2$$

$$\leq 4\cdot||w_t - w_t^*\otimes\mathbf{1}||^2 = 4\sum_{i=1}^{N}||w_t^i - w_t^*||^2 \tag{35}$$

where the third equality is from

$$\tilde{\delta}_{t,l}^{i}(w_t) - \tilde{\delta}_{t,l}^{i}(w_t^*)$$

$$= [A^{t_\text{gossip}}]_i\vec{\delta}_{t,l}(w_t) - [A^{t_\text{gossip}}]_i\vec{\delta}_{t,l}(w_t^*)$$

$$= [A^{t_\text{gossip}}]_i[\vec{\delta}_{t,l}(w_t) - \vec{\delta}_{t,l}(w_t^*)]$$

$$= [A^{t_\text{gossip}}]_i\begin{pmatrix}\phi^T(s_{t,l+1})(w_t^1 - w_t^*) - \phi^T(s_{t,l})(w_t^1 - w_t^*)\\\vdots\\\phi^T(s_{t,l+1})(w_t^N - w_t^*) - \phi^T(s_{t,l})(w_t^N - w_t^*)\end{pmatrix}$$

$$= [A^{t_\text{gossip}}]_i(\mathbf{w}_t - w_t^*\mathbf{1}^T)^T[\phi(s_{t,l+1}) - \phi(s_{t,l})].$$

For the second term in the RHS of equation 34, we have

$$||v_t^i(w_t^*) - h_t^i(w_t^*)||^2$$

$$= ||\frac{1}{B}\sum_{l=0}^{B-1}\tilde{\delta}_{t,l}^{i}(w_t^*)\cdot\psi_{t,l}^{i} - \frac{1}{B}\sum_{l=0}^{B-1}\delta_{t,l}(w_t^*)\cdot\psi_{t,l}^{i}||^2$$

$$= ||\frac{1}{B}\sum_{l=0}^{B-1}[\tilde{\delta}_{t,l}^{i}(w_t^*) - \delta_{t,l}(w_t^*)]\cdot\psi_{t,l}^{i}||^2$$

$$= ||\frac{1}{B}\sum_{l=0}^{B-1}\left([A^{t_\text{gossip}}]_i - \frac{1}{N}\mathbf{1}^T\right)\vec{\delta}_{t,l}(w_t^*)\cdot\psi_{t,l}^{i}||^2$$

$$\leq \max_{l\in\{0,\cdots,B-1\}}||\left([A^{t_\text{gossip}}]_i - \frac{1}{N}\mathbf{1}^T\right)\vec{\delta}_{t,l}(w_t^*)\cdot\psi_{t,l}^{i}||^2$$

$$\leq \max_{l\in\{0,\cdots,B-1\}}||[A^{t_\text{gossip}}]_i - \frac{1}{N}\mathbf{1}^T||^2\cdot||\vec{\delta}_{t,l}(w_t^*)||^2\cdot||\psi_{t,l}^{i}||^2$$

$$\leq \max_{l\in\{0,\cdots,B-1\}}N(2\frac{1+\eta^{-(N-1)}}{1-\eta^{N-1}}(1-\eta^{N-1})^{t_\text{gossip}+1})^2\cdot||\vec{\delta}_{t,l}(w_t^*)||^2$$

$$\leq 16N^2((1+\eta^{-(N-1)})(1-\eta^{N-1})^{t_\text{gossip}})^2(r_\text{max} + R_w)^2 = \kappa_3 N^2(1-\eta^{N-1})^{2t_\text{gossip}} \tag{36}$$

where $\kappa_3 = 16(1+\eta^{-(N-1)})^2(r_\text{max} + R_w)^2$. We note that the second equality is because

$$\tilde{\delta}_{t,l}^{i}(w_t^*) - \delta_{t,l}(w_t^*)$$

$$= [A^{t_\text{gossip}}]_i\vec{\delta}_{t,l}(w_t^*) - \frac{1}{N}\mathbf{1}^T\vec{\delta}_{t,l}(w_t^*)$$

$$= \left([A^{t_\text{gossip}}]_i - \frac{1}{N}\mathbf{1}^T\right)\vec{\delta}_{t,l}(w_t^*).$$

For the last inequality, we note that $\delta_{t,l}(w^*)$ is bounded because rewards and feature vectors are bounded, $\mu^i$ is bounded for constant step size and $w_t^*$ is bounded from the critic step. That is, for $j \in \mathcal{N}$ entry in $\vec{\delta}_{t,l}(w^*)$ by definition,

$$\delta_{t,l}^j(w^*) = r_{t,l}^j - \mu_{t,l}^j + [\phi(s_{t,l+1}) - \phi(s_{t,l})]^T w_t^*.$$

Hence, its 2-norm bound is

$$
\begin{aligned}
||\delta_{t,l}^j(w^*)|| &= ||r_{t,l}^j - \mu_{t,l}^j + [\phi(s_{t,l+1}) - \phi(s_{t,l})]^T w_t^*|| \\
&\leq ||r_{t,l}^j|| + ||\mu_{t,l}^j|| + ||\phi(s_{t,l+1}) - \phi(s_{t,l})|| \cdot ||w_t^*|| \\
&\leq r_{\max} + r_{\max} + [||\phi(s_{t,l+1})|| + ||\phi(s_{t,l})||] \cdot R_w \leq 2r_{\max} + 2R_w.
\end{aligned}
\tag{37}
$$

For the last term in equation 34, we have

$$
\begin{aligned}
&||g(w_t^*, \theta_t) - \nabla_\theta J(\theta_t)||^2 \\
&= ||\mathbb{E}_{d_{\theta_t}(s,a)}[\mathrm{Adv}_{w_t^*}(s,a)\psi_{\theta_t}(s,a)] - \mathbb{E}_{d_{\theta_t}(s,a)}[\mathrm{Adv}_{\theta_t}(s,a)\psi_{\theta_t}(s,a)]||^2 \\
&= ||\mathbb{E}_{d_{\theta_t}(s,a)}[(\mathrm{Adv}_{w_t^*}(s,a) - \mathrm{Adv}_{\theta_t}(s,a))\psi_{\theta_t}(s,a)]||^2 \\
&\leq (\mathbb{E}_{d_{\theta_t}(s,a)}[||(\mathrm{Adv}_{w_t^*}(s,a) - \mathrm{Adv}_{\theta_t}(s,a))\psi_{\theta_t}(s,a)||])^2 \\
&\leq (\mathbb{E}_{d_{\theta_t}(s,a)}[||\mathrm{Adv}_{w_t^*}(s,a) - \mathrm{Adv}_{\theta_t}(s,a)|| \cdot ||\psi_{\theta_t}(s,a)||])^2 \\
&\leq (\mathbb{E}_{d_{\theta_t}(s,a)}[|\mathrm{Adv}_{w_t^*}(s,a) - \mathrm{Adv}_{\theta_t}(s,a)|])^2 \\
&= (\mathbb{E}_{d_{\theta_t}(s,a)}[|\mathbb{E}[V_{w_t^*}(s')|s,a] - V_{w_t^*}(s) - \mathbb{E}[V_{\theta_t^*}(s')|s,a] + V_{\theta_t^*}(s)|])^2 \\
&\leq (\mathbb{E}_{d_{\theta_t}(s,a)}[|\mathbb{E}[V_{w_t^*}(s') - V_{\theta_t^*}(s')|s,a]| + |V_{w_t^*}(s) - V_{\theta_t^*}(s)|])^2 \\
&\leq (\mathbb{E}_{d_{\theta_t}(s,a)}[\mathbb{E}[|V_{w_t^*}(s') - V_{\theta_t^*}(s')||s,a] + |V_{w_t^*}(s) - V_{\theta_t^*}(s)|])^2 \\
&= (\mathbb{E}[|V_{w_t^*}(s) - V_{\theta_t^*}(s)|] + \mathbb{E}[|V_{w_t^*}(s) - V_{\theta_t^*}(s)|])^2 \\
&\leq 4(\mathbb{E}[|V_{w_t^*}(s) - V_{\theta_t^*}(s)|])^2 \leq 4\mathbb{E}[|V_{w_t^*}(s) - V_{\theta_t^*}(s)|^2] \leq 4\xi_{\mathrm{approx}}^{\mathrm{critic}}.
\end{aligned}
\tag{38}
$$

For each $i \in \mathcal{N}$, we have

$$
\begin{aligned}
&||h_t^i(w_t^*) - g^i(w_t^*, \theta_t^i)||^2 \\
&= \langle \frac{1}{B}\sum_{l_1=0}^{B-1}\delta_{t,l_1}(w_t^*)\psi_{\theta_t^i} - \mathbb{E}_{s,a}[\mathrm{Adv}_{w_t^*}(s,a)\psi_{\theta_t^i}^i(s,a^i)], \frac{1}{B}\sum_{l_2=0}^{B-1}\delta_{t,l_2}(w_t^*)\psi_{\theta_t^i} - \mathbb{E}_{s,a}[\mathrm{Adv}_{w_t^*}(s,a)\psi_{\theta_t^i}^i(s,a^i)]\rangle \\
&= \frac{1}{B^2}\sum_{l_1=0}^{B-1}\sum_{l_2=0}^{B-1}\langle\delta_{t,l_1}(w_t^*)\psi_{\theta_t^i} - \mathbb{E}_{s,a}[\mathrm{Adv}_{w_t^*}(s,a)\psi_{\theta_t^i}^i(s,a^i)], \delta_{t,l_2}(w_t^*)\psi_{\theta_t^i} - \mathbb{E}_{s,a}[\mathrm{Adv}_{w_t^*}(s,a)\psi_{\theta_t^i}^i(s,a^i)]\rangle \\
&= \frac{1}{B^2}\sum_{l=0}^{B-1}||\delta_{t,l}(w_t^*)\psi_{\theta_t^i} - \mathbb{E}_{s,a}[\mathrm{Adv}_{w_t^*}(s,a)\psi_{\theta_t^i}^i(s,a^i)]||^2 \\
&\quad + \frac{1}{B^2}\sum_{l_1\neq l_2}\langle\delta_{t,l_1}(w_t^*)\psi_{\theta_t^i} - \mathbb{E}_{s,a}[\mathrm{Adv}_{w_t^*}(s,a)\psi_{\theta_t^i}^i(s,a^i)], \delta_{t,l_2}(w_t^*)\psi_{\theta_t^i} - \mathbb{E}_{s,a}[\mathrm{Adv}_{w_t^*}(s,a)\psi_{t,l_2}^i(s,a^i)]\rangle.
\end{aligned}
$$

Taking expectation over the filtration $\mathcal{F}_t$, we have

$$
\begin{aligned}
&\mathbb{E}[||h_t^i(w_t^*) - g^i(w_t^*, \theta_t^i)||^2|\mathcal{F}_t] \\
&= \frac{1}{B^2}\sum_{l=0}^{B-1}\mathbb{E}\left[||\delta_{t,l}(w_t^*)\psi_{\theta_t^i} - \mathbb{E}_{s,a}[\mathrm{Adv}_{w_t^*}(s,a)\psi_{\theta_t^i}^i(s,a^i)]||^2|\mathcal{F}_t\right] \\
&\quad + \frac{1}{B^2}\sum_{l_1\neq l_2}\mathbb{E}\left[\langle\delta_{t,l_1}(w_t^*)\psi_{\theta_t^i} - \mathbb{E}_{s,a}[\mathrm{Adv}_{w_t^*}(s,a)\psi_{\theta_t^i}^i(s,a^i)], \delta_{t,l_2}(w_t^*)\psi_{\theta_t^i} - \mathbb{E}_{s,a}[A_{w_t^*}(s,a)\psi_{\theta_t^i}^i(s,a^i)]\rangle|\mathcal{F}_t\right] \\
&\leq \frac{16}{B}(r_{\max} + R_w)^2 \\
&\quad + \frac{1}{B^2}\sum_{l_1\neq l_2}\mathbb{E}\left[\langle\delta_{t,l_1}(w_t^*)\psi_{\theta_t^i} - \mathbb{E}_{s,a}[\mathrm{Adv}_{w_t^*}(s,a)\psi_{\theta_t^i}^i(s,a^i)], \delta_{t,l_2}(w_t^*)\psi_{\theta_t^i} - \mathbb{E}_{s,a}[\mathrm{Adv}_{w_t^*}(s,a)\psi_{\theta_t^i}^i(s,a^i)]\rangle|\mathcal{F}_t\right]
\end{aligned}
$$

where the inequality follows from triangle inequality and the facts that $|\delta_{t,l_1}(w_t^*)\psi_{\theta_t^i}| \leq 2r_{\max} + 2R_w$ and $|\mathbb{E}_{s,a}[\text{Adv}_{w_t^*}(s,a)\psi_{\theta_t^i}^i(s,a^i)]| = |\mathbb{E}[\delta_{t,l_1}(w_t^*)\psi_{\theta_t^i}]| \leq 2r_{\max} + 2R_w$. WLOG, for the following term, we suppose $l_1 < l_2$. Then we have

$$\mathbb{E}\left[\langle \delta_{t,l_1}(w_t^*)\psi_{\theta_t^i} - \mathbb{E}_{s,a}[\text{Adv}_{w_t^*}(s,a)\psi_{\theta_t^i}^i(s,a^i)], \delta_{t,l_2}(w_t^*)\psi_{\theta_t^i} - \mathbb{E}_{s,a}[\text{Adv}_{w_t^*}(s,a)\psi_{\theta_t^i}^i(s,a^i)]\rangle|\mathcal{F}_t\right]$$

$$=\mathbb{E}\left[\mathbb{E}\left[\langle \delta_{t,l_1}(w_t^*)\psi_{\theta_t^i} - \mathbb{E}_{s,a}[\text{Adv}_{w_t^*}(s,a)\psi_{\theta^i}^i(s,a^i)], \delta_{t,l_2}(w_t^*)\psi_{\theta_t^i} - \mathbb{E}_{s,a}[\text{Adv}_{w_t^*}(s,a)\psi_{\theta_t^i}^i(s,a^i)]\rangle|\mathcal{F}_{t,l_1}\right]|\mathcal{F}_t\right]$$

$$=\mathbb{E}\left[\langle \delta_{t,l_1}(w_t^*)\psi_{\theta_t^i} - \mathbb{E}_{s,a}[\text{Adv}_{w_t^*}(s,a)\psi_{\theta^i}^i(s,a^i)], \mathbb{E}\left[\delta_{t,l_2}(w_t^*)\psi_{\theta_t^i} - \mathbb{E}_{s,a}[\text{Adv}_{w_t^*}(s,a)\psi_{\theta_t^i}^i(s,a^i)]|\mathcal{F}_{t,l_1}\right]\rangle|\mathcal{F}_t\right]$$

$$=\mathbb{E}\left[\langle \delta_{t,l_1}(w_t^*)\psi_{\theta_t^i} - \mathbb{E}_{s,a}[\text{Adv}_{w_t^*}(s,a)\psi_{\theta^i}^i(s,a^i)], \mathbb{E}[\delta_{t,l_2}(w_t^*)\psi_{\theta_t^i}|\mathcal{F}_{t,l_1}] - \mathbb{E}_{s,a}[\text{Adv}_{w_t^*}(s,a)\psi_{\theta_t^i}^i(s,a^i)]\rangle|\mathcal{F}_t\right]$$

$$=\mathbb{E}\left[\langle \delta_{t,l_1}(w_t^*)\psi_{\theta_t^i} - \mathbb{E}_{s,a}[\text{Adv}_{w_t^*}(s,a)\psi_{\theta^i}^i(s,a^i)], \mathbb{E}[\text{Adv}_{w_t^*}(s_{t,l_2},a_{t,l_2})\psi_{\theta_t^i}|\mathcal{F}_{t,l_1}] - \mathbb{E}_{s,a}[\text{Adv}_{w_t^*}(s,a)\psi_{\theta_t^i}^i(s,a^i)]\rangle|\mathcal{F}_t\right]$$

$$\leq\mathbb{E}\left[||\delta_{t,l_1}(w_t^*)\psi_{\theta_t^i} - \mathbb{E}_{s,a}[\text{Adv}_{w_t^*}(s,a)\psi_{\theta^i}^i(s,a^i)]|| \cdot ||\mathbb{E}[\text{Adv}_{w_t^*}(s_{t,l_2},a_{t,l_2})\psi_{\theta_t^i}|\mathcal{F}_{t,l_1}] - \mathbb{E}_{s,a}[\text{Adv}_{w_t^*}(s,a)\psi_{\theta_t^i}^i(s,a^i)]|||\mathcal{F}_t\right]$$

$$\leq2(2r_{\max} + 2R_w)\mathbb{E}\left[||\mathbb{E}[\text{Adv}_{w_t^*}(s_{t,l_2},a_{t,l_2})\psi_{\theta_t^i}|\mathcal{F}_{t,l_1}] - \mathbb{E}_{s,a}[\text{Adv}_{w_t^*}(s,a)\psi_{\theta_t^i}^i(s,a^i)]|||\mathcal{F}_t\right]$$

$$\leq16(r_{\max} + R_w)^2\kappa\rho^{l_2-l_1}$$

where the last inequality follows from

$$||\mathbb{E}[\text{Adv}_{w_t^*}(s_{t,l_2},a_{t,l_2})\psi_{\theta_t^i}|\mathcal{F}_{t,l_1}] - \mathbb{E}_{s,a}[\text{Adv}_{w_t^*}(s,a)\psi_{\theta_t^i}^i(s,a^i)]||$$

$$=||\sum_{(s_{t,l_2},a_{t,l_2})}\text{Adv}_{w_t^*}(s_{t,l_2},a_{t,l_2})\psi_{\theta_t^i}(s_{t,l_2},a_{t,l_2})P(s_{t,l_2},a_{t,l_2}|\mathcal{F}_{t,l_1}) - \sum_{(s,a)}\text{Adv}_{w_t^*}(s,a)\psi_{\theta_t^i}^i(s,a)\nu_{\theta_t}(s,a)||$$

$$\leq\sum_{s,a}||\text{Adv}_{w_t^*}(s,a)\psi_{\theta_t^i}^i(s,a^i)|| \cdot |P^{l_2-l_1}(s,a|\mathcal{F}_{t,l_2}) - \nu_{\theta_t}(s,a)|$$

$$\leq2(2r_{\max} + 2R_w) \cdot ||P^{l_2-l_1}(s,a|\mathcal{F}_{t,l_2}) - \nu(s,a)||_{TV}$$

$$\leq4(r_{\max} + R_w)\kappa\rho^{l_2-l_1}.$$

Then, we have

$$\mathbb{E}[||h_t^i(w_t^*) - g^i(w_t^*,\theta_t^i)||^2|\mathcal{F}_t]$$

$$\leq\frac{1}{B^2}[16B(r_{\max} + R_w)^2 + 16(r_{\max} + R_w)^2\kappa\sum_{l_2\neq l_1}\rho^{l_2-l_1}]$$

$$\leq\frac{1}{B^2}[16B(r_{\max} + R_w)^2 + \frac{32(r_{\max} + R_w)^2\kappa\rho B}{1-\rho}]$$

$$\leq\frac{16(r_{\max} + R_w)^2[1+(2\kappa-1)\rho]}{B(1-\rho)}. \tag{39}$$

Then, we have

$$\mathbb{E}[||v_t(w_t) - \nabla_\theta J(\theta_t)||^2] \leq24N\sum_{i=1}^{N}||w_t^i - w_t^*||^2 + 6\kappa_3 N^3(1-\eta^{N-1})^{2t_{\text{gossip}}}$$

$$+ 24\zeta_{\text{approx}}^{\text{critic}} + \frac{96(r_{\max} + R_w)^2[1+(2\kappa-1)\rho]}{B(1-\rho)}N \tag{40}$$

As a result, substituting equation equation 40 into equation 33 and taking expectation over $\mathcal{F}_t$ on both sides, we have

$$(\frac{1}{2}\alpha - L_J\alpha^2)\mathbb{E}[||\nabla_\theta J(\theta_t)||^2]$$

$$\leq \mathbb{E}[J(\theta_{t+1})] - \mathbb{E}[J(\theta_t)] + (\frac{1}{2}\alpha + L_J\alpha^2)\left(24N\sum_{i=1}^{N}||w_t^i - w_t^*||^2\right.$$

$$+6\kappa_3 N^3(1-\eta^{N-1})^{2t_{\text{gossip}}} + 24\xi_{\text{approx}}^{\text{critic}} + 96\frac{(r_{\max}+R_w)^2[1+(2\kappa-1)\rho]}{B(1-\rho)}N\Bigg). \tag{41}$$

By considering step size $\alpha = \frac{1}{4L_J}$, and dividing both sides of previous equation by $\frac{1}{16L_J}$, we have

$$\mathbb{E}[||\nabla_\theta J(\theta_t)||^2] \leq 16L_J(\mathbb{E}[J(\theta_{t+1})] - \mathbb{E}[J(\theta_t)]) + 72N\sum_{i=1}^{N}||w_t^i - w_t^*||^2$$

$$+ 18\kappa_3 N^3(1-\eta^{N-1})^{2t_{\text{gossip}}} + 72\xi_{\text{approx}}^{\text{critic}} + 288N\frac{(r_{\max}+R_w)^2[1+(2\kappa-1)\rho]}{B(1-\rho)}.$$

$$\tag{42}$$

Let $\hat{T}$ be a random variable that takes value uniformly among $\{1, \cdots, T\}$. Taking summation over $t = \{1, \cdots, T\}$ and dividing by $T$, we have

$$\mathbb{E}[||\nabla_\theta J(\theta_{\hat{T}})||^2] = \frac{1}{T}\sum_{t=1}^{T}\mathbb{E}[||\nabla_\theta J(\theta_t)||^2]$$

$$\leq \frac{16L_J(\mathbb{E}[J(\theta_T)] - \mathbb{E}[J(\theta_0)])}{T} + 72N\frac{\sum_{t=1}^{T}\sum_{i=1}^{N}||w_t^i - w_t^*||^2}{T}$$

$$+ 18\kappa_3 N^3(1-\eta^{N-1})^{2t_{\text{gossip}}} + 72\xi_{\text{approx}}^{\text{critic}} + 288\frac{(r_{\max}+R_w)^2[1+(2\kappa-1)\rho]}{B(1-\rho)}N$$

$$\leq \frac{16L_J\mathbb{E}[J(\theta_T)]}{T} + 72N\frac{\sum_{i=1}^{N}||w_t^i - w_t^*||^2}{T}$$

$$+ 18\kappa_3 N^3(1-\eta^{N-1})^{2t_{\text{gossip}}} + 72\xi_{\text{approx}}^{\text{critic}} + 288\frac{(r_{\max}+R_w)^2[1+(2\kappa-1)\rho]}{B(1-\rho)}N.$$

