# OpenReview forum: "Finite-Time Convergence and Sample Complexity of Multi-Agent Actor-Critic Reinforcement Learning with Average Reward"
_ICLR.cc/2022/Conference — ICLR 2022 Spotlight_

### Official Review · Reviewer_AYYd · 2021-11-03

**Correctness:** 4
**Technical Novelty And Significance:** 3
**Empirical Novelty And Significance:** 2
**Recommendation:** 6
**Confidence:** 3

**Main Review:**

- The paper is in general well written and well motivated. The ideas are clearly explained and presented in an organized way.
- The log(N^5/\epsilon) term in the complexity appears odd. It seems that that log(N/\epsilon) is enough.
- The assumption on observable global-state and unobservable global action should be better justified. It would be helpful to provide an example for it..
- Assumption 1 needs more explanation. It seems to assume that the optimal policy will always utilize all actions with non-zero probability? Usually it is more common that a deterministic policy exists for many problems. Please elaborate.
- There is a typo in Assumption 5. “bdd” should be “bounded”.
- The last sentence in Assumption 5 should be better explained. What does the condition imply?
 - Theorem 2 seems to indicate that a smaller beta gives a better result? Can the authors provide some intuition?
- The paper claims that the current work is still an improvement of Zhang et al 2018. Can this be made more specified?
- In the experimental section, the reviewer wonders whether schemes from the discounted case with gamma->1 can be compared here? Intuitively, they should also give reasonable performance.
- Any intuition for the fact that the empirical sample complexity does not scale with N? Could this be because of the special topology used?
- It might be helpful to strengthen the experimental section. The current setup is a bit limited.

**Summary Of The Paper:**

The paper studies the finite-time convergence and sample complexity of MARL with average reward, focusing on actor-critic algorithm. It proposes a mini-batch Markovian sampled algorithm and obtain its sample complexity.


**Summary Of The Review:**

The paper is well written and the topic is timely and of interest. Some assumptions/statements require more elaboration and the experiments should be strengthened.

---

> ### Author Response · Authors · 2021-11-22
> **Response to Reviewer AYYd**
>
> We sincerely thank the reviewer's constructive comments and valuable insights, which help improve the quality of our work significantly. We have carefully revised our paper according to your comments and suggestions. Please see our revised submission, where we have highlighted all major changes in **Red color**. Please also see our point-to-point responses as follows:
>
> > **Your Comment:** 1. The paper is in general well written and well motivated. The ideas are clearly explained and presented in an organized way. The $\log(N^5/\epsilon)$ term in the complexity appears odd. It seems that that $\log(N/\epsilon)$ is enough.
>
> **Our Response:** Yes, we agree that the original expression is odd and it can indeed be simplified to $\log(N/\epsilon)$.
> Thanks for pointing this out. We have revised the order-wise expression in the paper.
>
> > **Your Comment:** 2. The assumption on observable global-state and unobservable global action should be better justified. It would be helpful to provide an example for it.
>
> **Our Response:** Thanks for your comment. Here, we would like to further provide some motivating examples for our assumptions in MARL. The setting with unobservable joint actions and observable global states is motivated by the following examples in multi-agent systems:
>
> * *Example 1:* In autonomous driving (Yu et al., 19), each vehicle/driver can only detect/observe the actions of its surrounding vehicles that are within its communication/visible range (e.g., action of vehicle $i$, denoted by $a_i$, could be turn left, turn right, stop, start, etc.).
> However, the joint action of all vehicles (e.g., city-wide) cannot be observed since most of them are beyond the observable range of vehicle $i$. However, the global state $s_t$ at time $t$ may still be observable in this application, e.g., congestion level of all roads can be observed by Google Maps, etc.
>
> * *Example 2:* In the control of carrier-sense multiple access(CSMA) protocol for wireless networks (e.g., medium access control for WiFi networks), the nodes (i.e., agents) transmit through a shared wireless channel/medium. In this example, the state of the shared transmission channel in time $t$, denoted by $s_t$, is the global state (e.g., idle or occupied), which each agent can detect by sensing (listening ongoing traffic if any). The CSMA control policy we consider is the so-called $p$-persistent CSMA: when an agent $i$ detects the medium is idle, it would transmit with probability $p_i \in [0,1]$, where transmitting or not are the actions. Typically, in the fully distributed CSMA protocol (see Ref. [H]), the agents do not have the knowledge of the other nodes' transmit probabilities (or actions $a_j$, $j\ne i$). Therefore, in this $p$-persistent CSMA-based wireless networks, the global state is observable, but the joint actions are not.
>
> [H] Moon, Sangwoo, et al. "Neuro-DCF: Design of Wireless MAC via Multi-Agent Reinforcement Learning Approach." in Proceedings of the Twenty-second International Symposium on Theory, Algorithmic Foundations, and Protocol Design for Mobile Networks and Mobile Computing. 2021.
>
> > **Your Comment:** 3. Assumption 1 needs more explanation. It seems to assume that the optimal policy will always utilize all actions with non-zero probability? Usually it is more common that a deterministic policy exists for many problems. Please elaborate.
>
> **Our Response:** Thanks for pointing out this issue. This is a typo, which should be corrected as non-negative probability, i.e., $\pi(a^{i}|s)\ge 0$ for all $i, a^{i}$ and $s$.
>
> > **Your Comment:** 4. There is a typo in Assumption 5. “bdd” should be “bounded”. The last sentence in Assumption 5 should be better explained. What does the condition imply?
>
> **Our Response:** Thank you for catching this typo. We have fixed the typo in this revision. Here, we would further clarify the last sentence of Assumption 5 in the original submission. Note that since we have rewritten the original Assumption 2 as a lemma implied by Assumption 1 (as suggested by other reviewers), the original Assumption 5 now becomes Assumption 4 in this revision. First, we remark that the assumption $\Phi u \neq \mathbf{1}$ is mild and can be satisfied easily in practice. This is because, in many MARL systems, it often holds that the size of the state space is greater than the feature mapping dimension, i.e., $|\mathcal{S}| > K$. Thus, the matrix $\Phi\in \mathcal{R}^{|\mathcal{S}|\times K}$, $|\mathcal{S}|>K$ is a tall and skinny matrix, which implies that, mathematically, the linear equation system $\Phi u=\mathbf{1}$ is overdetermined. Thus, there is no solution under a proper choice of the feature matrix $\Phi$.
> Moreover, this assumption ensures  $A_{\pi_{\theta}}$ defined in Eq.~(1) to be invertible (see the following reference [C]).
> This assumption in turn implies that the policy evaluation has a unique stable condition (please also see [Zhang et al., 2018]).
>
> **Please continue to see our next response below.**

---

> > ### Author Response · Authors · 2021-11-22
> > **Response to Reviewer AYYd (Continued)**
> >
> > > **Your Comment:** 5. Theorem 2 seems to indicate that a smaller beta gives a better result? Can the authors provide some intuition?
> >
> > **Our Response:** Thanks for your question. We want to clarify that Theorem 2 does not necessarily imply a smaller $\beta$-value is better.
> > Theorem 2 only implies that a smaller $\beta$-value within a certain range *may* work well. In fact, the impact of the $\beta$-value in Theorem 2 is indefinite. To see this, from Eq. (8), we have that the sum of the second and third terms shrink as $\beta$ becomes smaller.
> > However, the fourth term becomes larger when the $\beta$-value decreases. Therefore, how the right-hand-side of Eq. (8) would change as the value of $\beta$ decreases really depends on which term is dominant (the sum of the second and third terms, or the forth term).
> >
> > > **Your Comment:** 6. The paper claims that the current work is still an improvement of Zhang et al 2018. Can this be made more specified?
> >
> > **Our Response:** Thanks for your suggestions. Here, we would like to further explain our claim regarding the improvement of (Zhang et al. 2018). Our claimed improvement is in terms of the ratio of sample size over communication round. As indicated in Theorem 3 in our paper, the sample complexity is $O(\epsilon^{-2}\log \epsilon^{-1})$ and the communication complexity is $O(\epsilon^{-1}\log \epsilon^{-1})$.
> > Hence, the sample size to communication round ratio is of $O(\epsilon^{-1})$, while the corresponding ratio in (Zhang et al 2018) is 1.
> > However, this improvement over (Zhang et al. 2018) does not imply (Zhang et al. 2018) is inferior, since the batching modification of (Zhang et al. 2018) can also improve this ratio to $B$, where $B$ is the batch size. By choosing the batch size as $B=1/\epsilon$, the work in (Zhang et al. 2018) can achieve the same ratio as ours.
> >
> > > **Your Comment:** 7. In the experimental section, the reviewer wonders whether schemes from the discounted case with gamma -> 1 can be compared here? Intuitively, they should also give reasonable performance.
> >
> > **Our Response:** Thanks for the suggestion. Following your suggestion, we have conducted a discounted TD-sharing counterpart algorithm to illustrate the results numerically. Please see Figure 2(d) in Section A.2. Our numerical results show the objective numerical values of the $\gamma$-discounted setting approach that of the average reward setting in general when $\gamma$ approaches 1.
> > However, we can see average setting value converges to a significantly higher value compared to the $\gamma$-discounted setting even with a large $\gamma$-value. More importantly, one advantage of the average reward setting is that, with more samples, the policies can potentially keep updating and so can the objective value. From Figure 2(d), we can see that as the number of samples increases, the objective value of the average reward setting continues to evolve, which means that the policies are keep updating. However, for the discounted reward case, the extra sample does not affect the objective value.
> >
> > > **Your Comment:** 8. Any intuition for the fact that the empirical sample complexity does not scale with $N$? Could this be because of the special topology used?
> >
> > **Our Response:** At this point, we do not have a strong intuitive explanation as to why the empirical sample complexity does not scale with the system size $N$. We conjecture that, in addition to the size of the network, network topology could also play an important role.
> > To further understand the effect of network topology on the performance, we have conducted additional simulations. We can see from Figure 3 that different network topologies have similar but different impacts on reward performance. Moreover, all results show performance improvement over the baseline value of 2. That the numerical insensitivity of the reward performance with respect to the network size indicates that our theoretical complexity bounds may be further sharpened. The potential improvement, as well as the effect of network topologies, are very interesting topics in our future studies. Thus, we thank the reviewer for this insightful question.
> >
> > > **Your Comment:** 9. It might be helpful to strengthen the experimental section. The current setup is a bit limited.
> >
> > **Our Response:** Thank you for your suggestion. In this revision, we have added new numerical experiment comparison results with variants of the classical MARL algorithm in (Zhang et al. 2018), including both constant step-size and batching sampling modifications.
> > As shown in our new numerical experiments, our algorithm outperforms the classical MARL algorithm and its variants with different system sizes. This suggests that, empirically, our proposed double-loop structure combined with the consensus TD sharing approach is beneficial. Please see Section A.2 for detailed comparisons.

---

> > > ### Comment · Reviewer_AYYd · 2021-12-07
> > > **Thanks for the detailed responses**
> > >
> > > The reviewer would like to thank the authors for the careful and detailed responses. The concerns are well addressed.

---

### Official Review · Reviewer_avem · 2021-11-05

**Correctness:** 3
**Technical Novelty And Significance:** 2
**Empirical Novelty And Significance:** 1
**Recommendation:** 6
**Confidence:** 3

**Main Review:**

In general, the paper is well-written, well-motivated, and the literature review is very complete. I enjoyed reading the paper. The detailed comments are as follows:

1. It would be better if the authors could explicitly summarize the difference, and mainly, the technical challenges of the analyses for the decentralized average reward setting, compared to the discounted setting and the centralized setting, in the introduction of the paper.
2. I wonder what is the difference of the Algorithm 2 in the paper from the Algorithm 2 in Zhang et al., 2018? In the Algorithm 2 therin, the "global average reward" $\bar r$ needs to be estimated separately, in order to obtain the right TD error estimate. It seems a bit surprising to me that this paper can do it using only $r^i$. A detailed comparison and intuition would be necessary.
3. In the statement of Theorem 3, how large is the eps^{critic}_{approx? Would the upper bound for stationarity be vacuous? Some discussions are necessary.
4. It would be better if the simulation section could include more complicated and realistic settings. The current empirical contribution is a bit limited.
5. Some typos: On the top of page 2: "without join action" should be "without joint action"; Assumption 5. “bdd” should be “bounded”.

**Summary Of The Paper:**

This paper establishes the first finite-time convergence result of the actor-critic algorithm for fully decentralized multi-agent reinforcement learning (MARL) problems with average reward. It focuses on the practical setting where the rewards and actions of each agent are only known to itself, and the knowledge of joint actions of the agents is not assumed. The established finite-sample complexity matches that of the state-of-the-art single-agent actor-critic algorithms.

**Summary Of The Review:**

The paper has made valid contribution to the area of multi-agent RL, and is well-written. As the empirical contribution is a bit limited, so I would view the main contribution to be theoretical. However, some detailed comparison with the most related works is needed, before justifying the novelty and significance of the theoretical contribution.

---

> ### Author Response · Authors · 2021-11-22
> **Response to Reviewer avem**
>
> We sincerely thank the reviewer's constructive comments and valuable insights, which help improve the quality of our work significantly. We have carefully revised our paper according to your comments and suggestions. Please see our revised submission, where we have highlighted all major changes in **Red color**. Please also see our point-to-point responses as follows:
>
> > **Your Comment:** 1. It would be better if the authors could explicitly summarize the difference, and mainly, the technical challenges of the analyses for the decentralized average reward setting, compared to the discounted setting and the centralized setting, in the introduction of the paper.
>
> **Our Response:** Thanks for your comments. Here, we summarize the differences and challenges of the analyses of our settings as follows:
> * *Differences between the Average and Discounted Reward Settings:* The average and discounted reward settings differ from each other in the following key aspects:
>     * *Uniqueness of the solution:* In the discounted reward setting, due to the existence of a discounting factor with a value strictly less than one, the corresponding Bellman equation for a given policy exhibits a "contraction" property. Unfortunately, in the average reward setting, this is no such a discounting factor, which may cause the Bellman equation to have infinitely many solutions. In the average reward setting, with an additional constraint $d_{\theta}^{T} V_{\theta}=0$, where $d_{\theta}$ is the state distribution vector and $V_{\theta}$ is the state value function vector under policy $\pi_{\theta}$, the solution to the Bellman equation becomes unique, which is referred to as the basic differential value function (Tsitsiklis \& Van Roy,. 1999). The basic differential value function means that, rather than the value at each state, what really matters is the difference between value functions at different states. Although the discounted total reward setting captures the important aspect of diminished return in the future, it may not be appropriate for many other applications where the *long-term average reward* is of interest. For example, in the optimization of distributed communication networks with MARL, the typical and natural performance metrics are long-term average throughput or latency in the steady-state, see for example [H].
>
>     * *Truncation of the horizon:* Note that the objective function in the discounted reward setting is geometrically weighted, as opposed to the uniform weighted form in the average reward setting. This entails an effective horizon of $O(\frac{\log^{\epsilon^{-1}}}{1-\gamma})$ for the discounted reward setting to reach an $\epsilon$-approximate stationary point (see reference [E]). Therefore, many well-known techniques in the discounted reward setting cannot be carried over to the average reward setting even in single-agent reinforcement learning. For multi-agent reinforcement learning, these differences continue to hold.
>
> * *Differences between the Decentralized and Centralized Settings:* The difference between the decentralized and centralized settings mainly stems from the different information structures. Specifically, in the case of the centralized setting, usually, a server/controller knows all the information gathered by the agents, hence the full knowledge of the global states, joint actions, and rewards from agents, all of which can be utilized to design efficient policies for the agents. In the decentralized setting, however, due to the lack of such a centralized server/controller, the agents can only leverage the local information and exchange information with their immediate neighbors in order to design the policies. This means that the agents in the decentralized setting need to design a comparable policy to the centralized case with less information, thus potentially larger communication costs.
>
> [F] Kakade, S.M., 2003. On the sample complexity of reinforcement learning. The University of London, University College London (United Kingdom).
>
> [H] Moon, Sangwoo, et al. "Neuro-DCF: Design of Wireless MAC via Multi-Agent Reinforcement Learning Approach." in Proceedings of the Twenty-second International Symposium on Theory, Algorithmic Foundations, and Protocol Design for Mobile Networks and Mobile Computing. 2021.
>
> > **Your Comment:** 2. I wonder what is the difference of the Algorithm 2 in the paper from the Algorithm 2 in Zhang et al., 2018? In the Algorithm 2 therin, the "global average reward" r¯  needs to be estimated separately, in order to obtain the right TD error estimate. It seems a bit surprising to me that this paper can do it using only ri. A detailed comparison and intuition would be necessary.
>
> **Please continue to see our next response below.**

---

> > ### Author Response · Authors · 2021-11-22
> > **Response to Reviewer avem (Continued)**
> >
> > **Our Response:** Thanks for your questions. Here, we would like to further clarify the difference between our work and (Zhang et al. 2018).
> > In (Zhang et al., 2018), the setting allows the observability of both the global state and joint action. In order to compute the global TD error, they estimate the global average reward. By contrast, in our approach, since the global TD error is the average of the local TD errors, we directly estimate global TD error by a consensus procedure. Specifically, from Line 12-17 of Algorithm 2, there is a consensus process with $t_{\text{gossip}}$ iterations. The trade-off of our approach is that, in order to reach a good estimate of the global TD error for each agent, the consensus process needs to be conducted  $t_{\text{gossip}}$ rounds. We note that, thanks to the use of our proposed double-loop structure with consensus-based TD-error estimation, we are able to achieve finite-time convergence rate result as opposed to the asymptotic convergence result in (Zhang et al. 2018). To our knowledge, this is new in the literature.
> >
> > > **Your Comment:** 3. In the statement of Theorem 3, how large is the $\epsilon^{critic}_{approx}$? Would the upper bound for stationarity be vacuous? Some discussions are necessary.
> >
> > **Our Response:** Thanks for your questions. Here, we would like to provide further discussions on the range of $\epsilon^{critic}_{approx}$ and the upper bound:
> >
> > * In our paper, we assume the error term $\epsilon^{critic}_{approx}$ is bounded and sufficiently small.
> > Such a term has also been commonly assumed and has appeared in the previous studies of actor-critic algorithm, e.g., (Qiu et al., 2019) and (Xu et al., 2020).
> >
> > * For a given policy $\theta$, this linear approximation error term depends on how well the ground truth of the value functions coincides with the linear space spanned by the feature $\Phi$ defined in Assumption 5. If these two match sufficiently well, then the approximation error term $\epsilon^{critic}_{approx}$ would be small; otherwise, the error term could be large. From a technical perspective, such a linear approximation is relatively easier to analyze compared to nonlinear or general function approximations.
> >
> > > **Your Comment:** 4. It would be better if the simulation section could include more complicated and realistic settings. The current empirical contribution is a bit limited.
> >
> > **Our Response:** Thanks for your suggestions. Although our major contributions in this paper are more on the theoretical design and analysis of the proposed MARL algorithm, we do agree that conducting experiments with more complicated realistic settings is important.
> > Following your suggestions, in this paper, we have added new empirical comparison results with the variants of the classical MARL algorithms in (Zhang et al. 2018), including both constant step-size and batch sampling modifications. Our new experimental results show that our proposed algorithm empirically outperforms the classical MARL algorithm and its variants with different system sizes.
> > This suggests that, empirically, the double-loop structure combined with the consensus TD sharing approach might be beneficial. See Section A.2 for detailed comparisons.
> >
> > > **Your Comment:** 5. Some typos: On the top of page 2: "without join action" should be "without joint action"; Assumption 5. “bdd” should be “bounded”.
> >
> > **Our Response:** Thanks for pointing out those typos. We have fixed these typos in this revision.

---

> ### Comment · Reviewer_avem · 2021-12-06
> **Thanks for the detailed response**
>
> I would like to thank the authors for the detailed response. I am happy with it, and have raised my score accordingly.

---

### Official Review · Reviewer_N673 · 2021-11-05

**Correctness:** 4
**Technical Novelty And Significance:** 3
**Empirical Novelty And Significance:** 2
**Recommendation:** 8
**Confidence:** 3

**Main Review:**

Strength
Only local action is observed, which is an improvement from [Zheng et al 2018].
Finite-time convergence is provided which matches those bounds in single-agent AC.

Weakness
1. The time index is a bit confusing. In Algorithm 1 (TD subroutine), there is a $s_{k,\tau}$ state sequence. In the outer loop, there is a $s_{t,\tau}$ sequence. This is quite confusing: do these states in the subroutine and the main Algorithm 2 form a single trajectory, or these are multiple trajectories?
2. In the TD subroutine, many TD errors are computed. Why can’t such TD errors be used to estimate the advantage function, as opposed to the approach in the paper (Line 1-10 in Algorithm 2) where many new TD errors are sampled?
3. The benefit of Batch is not sufficiently discussed. For example, how does the batch algorithm compare with the “non-batch” algorithm, where one update is made after each sample?
4. While $t_{gossip}$ scales with $1/\epsilon$, each time a $B$-length vector is communicated, and $B$ also scales with $1/\epsilon$. So the overall communication is somehow more than just $t_{gossip}$, and this should be discussed.
5. This simulation scope seems too limited (in the standard of ICLR).


**Summary Of The Paper:**

This paper studies a networked MARL problem based on the model in [Zhang et al 2018], where each agent can observe the global state, take local action and observe local rewards. The key difference in setting from [Zhang et al 2018] is that [Zhang et al 2018] assume the global action can be observed, but in this paper, only local action is known to each agent. To deal with this, an additional consensus loop is added to estimate the average TD error, which can be used to estimate the advantage function. Further, compared to [Zhang et al 2018], a finite time error bound is provided.


**Summary Of The Review:**

Overall this is an interesting generalization from [Zhang et al 2018], but I feel many points are worth more elaboration and discussion. More simulation is needed.

---

> ### Author Response · Authors · 2021-11-22
> **Response to Reviewer N673**
>
> We sincerely thank the reviewer's constructive comments and valuable insights, which help improve the quality of our work significantly. We have carefully revised our paper according to your comments and suggestions. Please see our revised submission, where we have highlighted all major changes in **Red color**. Please also see our point-to-point responses as follows:
>
> > **Your Comment:** 1. The time index is a bit confusing. In Algorithm 1 (TD subroutine), there is a $s_{k,\tau}$ state sequence. In the outer loop, there is a $s_{t,\tau}$ sequence. This is quite confusing: do these states in the subroutine and the main Algorithm 2 form a single trajectory, or these are multiple trajectories?
>
> **Our Response:** Thanks for pointing out this confusion. They form a single trajectory. We have replaced the second index of the sample in the actor step, i.e., Algorithm 2, into a different notation.
>
> > **Your Comment:** 2. In the TD subroutine, many TD errors are computed. Why can’t such TD errors be used to estimate the advantage function, as opposed to the approach in the paper (Line 1-10 in Algorithm 2) where many new TD errors are sampled?
>
> **Our Response:** Thank you for your insightful comment. This is a valid point since the suggested approach may reduce sample complexity, and we do plan to consider the possibility of such an approach in our future work. In our current paper, it is important to resample and compute the TD errors for the new samples. This is because in the TD subroutines, the main job is to get a good estimate of the value function approximation parameter. However, if one reuses the samples from the TD subroutine, it would generate inaccuracy in the actor update since, from Line 8 in the TD subroutine, the $w^{i}_k$ parameters are different for different iteration $k$. Specifically, the local TD errors would be biased, hence the global TD error, i.e. the average of the TD errors, would also be biased. In the end, the actor parameter update will be biased.(See Line 8 and 12-21 in Algorithm 1). As a result, it complicates the analysis in the actor step. Thanks to the use of the double-loop structure, we can easily avoid such complications.
>
> > **Your Comment:** 3. The benefit of Batch is not sufficiently discussed. For example, how does the batch algorithm compare with the “non-batch” algorithm, where one update is made after each sample?
>
> **Our Response:** Thanks for your suggestions. Here, we would like to further explain the benefits of batching as follows.
> On one hand, the benefit of the batch size (denoted by $M$) in the critic step is that, with larger batch size, one can obtain a smaller critic step error. More precisely, we can see that the last term in Eq. (8) of Theorem 2 will be smaller with a larger batch size $M$. Also, the benefit of the batch size (denoted by $B$) in the actor step is that it can also similarly decrease the error term in the actor convergence (see last term in Eq. (9) of Theorem 3). On the other hand, the local TD errors can be used as a $B$-length vector in the actor step to reach consensus, which reduces the number of communication rounds as compared to the non-batch approach.
>
> > **Your Comment:** 4. While $t_{\text{gossip}}$ scales with $1/\epsilon$, each time a B-length vector is communicated, and B also scales with $1/\epsilon$. So the overall communication is somehow more than just $t_{\text{gossip}}$, and this should be discussed.
>
> **Our Response:** Thanks for your comments. We do agree that $t_{\text{gossip}}$ indeed scales with $\log 1/\epsilon$ as indicated in the Theorem 3, and $B$ also scales with $1/\epsilon$. It is indeed true that the amount of information (in terms of bits) cannot be simply be defined by $t_{\text{gossip}}$. However, for the purpose of direct comparisons, we follow the standard definition of communication complexity in the literature, which is widely adopted in the literature (please also see [D][E] as examples). Here, the overall communication cost is measured by the number of communication rounds rather than the size of bits transmitted over the network. But we agree that highlighting the amount of communicated information in terms of bits is important, and we have added this discussion in the revision.
>
> [D] Chen, T., Giannakis, G.B., Sun, T. and Yin, W., 2018. LAG: Lazily aggregated gradient for communication-efficient distributed learning. arXiv preprint arXiv:1805.09965.
>
> [E] Zhang, X., Liu, J., Zhu, Z. and Bentley, E.S., 2019, April. Compressed distributed gradient descent: Communication-efficient consensus over networks. In Proceedings of IEEE INFOCOM 2019-IEEE Conference on Computer Communications, pp. 2431-2439.
>
> > **Your Comment:** 5. This simulation scope seems too limited (in the standard of ICLR).
>
> **Please continue to see our next response below.**

---

> > ### Author Response · Authors · 2021-11-22
> > **Response to Reviewer N673 (Continued)**
> >
> > **Our Response:** Thanks for your comments. In this revision, we have added additional empirical comparison results with the several variants of the classical MARL algorithms in (Zhang et al. 2018), including both constant step-sizes and batch sampling modifications.
> > Our new experimental results show that our algorithm empirically outperforms the classical MARL algorithm and its variants with different system sizes. See Section A.2 for detailed comparisons.

---

> > > ### Comment · Reviewer_N673 · 2021-11-24
> > > **Good response!**
> > >
> > > Good response!
> > >
> > > All my concerns have been clarified and addressed. I have revised the score accordingly.

---

### Official Review · Reviewer_F2da · 2021-11-07

**Correctness:** 4
**Technical Novelty And Significance:** 3
**Empirical Novelty And Significance:** 3
**Recommendation:** 8
**Confidence:** 4

**Main Review:**

[Reasons to accept]
This paper makes some relatively non-trivial progress on top of (Zhang et al., 2018). In particular, while (Zhang et al., 2018) requires that each agent observes the joint actions of all agents, this paper only requires each agent to observe their own private action. In addition, this paper also derives the first non-asymptotic bound for fully decentralized MARL on average reward MDPs. Thirdly, some numerical experiments are also provided to show that the modifications in this paper actually lead to improvement over the algorithm in (Zhang et al., 2018).

[Questions and concerns]
However, there are also some issues in this paper regarding assumptions, literature comparisons and numerical experiments that the authors need to address. See below for the details.
1. The comparison of the results with (Zhang et al., 2018) may not be sufficiently fair. In particular, the assumptions made in that paper seems to be different from the current paper (e.g., Assumption 7 seems not to be needed in (Zhang et al., 2018). I would suggest the authors to make it clearer in Section 3.2 which assumptions are different from (Zhang et al., 2018), why they are essential, and how they help obtaining the improved results (like the non-asymptotic results) in the current paper. On a related point, the clarity of Section 3.2 also needs some improvement. For example, Assumption 2 is implied by Assumption 1, and so should not be stated as a separate assumption. Also, the notation of consensus weight matrix, $A_{\pi_{\theta}}$ and advantage functions all use $A$, which might cause some confusion.
2. The authors should also compare the results with the recent global optimality literature of actor-critic methods, such as [A,B]. Are there any essential challenges in obtaining global optimality bounds using the techniques introduced in papers such as [A,B]? Also, in [A], it is mentioned that from both theoretical and practical perspectives, bi-level actor-critic methods which first update the critic to sufficient accuracies as in the current paper are typically seldom adopted in practice. However, the numerical results in this paper seem to suggest that the bi-level/mini-batch updates might indeed be beneficial. Some more discussion and numerical experiments are needed to validate this (see the next point for more comments).
3. I have some questions for the numerical experiments. The authors mentioned that constant step-sizes are key to the improvement over (Zhang et al., 2018). Then what if using the same constant step-sizes for the classical MARL algorithm in (Zhang et al., 2018)? And what about updating classical MARL in a mini-batch critic/bi-level manner as in this paper? I think providing these comparisons are important to understand what are essential in the modifications on top of the classical MARL algorithm. Otherwise, it gives me an impression that allowing for observing global actions is impeding the performance, which is counterintuitive.

There are also some minor suggestions, as detailed below:
1. In Assumption 5, why is $\Phi u \neq 1$ for any $u$ reasonable? I understand that this is a classical assumption, but still some explanations would be helpful.
2. Some more comments on what the quantity $A_{\pi_{\theta}}$ in (1) is would be helpful.
3. On page 6, 2) The Actor Step, “where in our model” might better be “whereas in our model”.
4. On page 7, what is “the weight information of other agents”?
5. It would be helpful to refer to the x-axis in the plots as “number of samples” instead of “sample complexity”. And just to double check, does one sample mean tuple of $(s,a,r,s’)$?

[A] Fu, Zuyue, Zhuoran Yang, and Zhaoran Wang. "Single-timescale actor-critic provably finds globally optimal policy." arXiv preprint arXiv:2008.00483 (2020).

[B] Yang, Zhuoran, Yongxin Chen, Mingyi Hong, and Zhaoran Wang. "On the global convergence of actor-critic: A case for linear quadratic regulator with ergodic cost." arXiv preprint arXiv:1907.06246 (2019).


**Summary Of The Paper:**

This paper considers cooperative multi-agent reinforcement learning (MARL) for average reward MDPs with fully decentralized actor-critic methods. In particular, the authors make some progress on top of existing works in this direction, and in particular (Zhang et al., 2018). More precisely, the authors remove the assumption in (Zhang et al., 2018) that the joint actions are observable to all agents, and propose to modify the actor updates with mini-batch TD sharing to accommodate the scenario where each agent only observes its own action. The authors then establish a finite-sample bound for the proposed algorithm in terms of convergence to stationary points under linear value function approximation. Numerical experiments are also provided to showcase the benefits of the modifications over the algorithm in (Zhang et al., 2018).

**Summary Of The Review:**

The theoretical contribution of this paper on top of (Zhang et al., 2018) is good. However, there also are some issues in terms of the assumptions, comparisons with the global convergence of actor-critic methods literature and numerical experiments that need to be addressed.

---

> ### Author Response · Authors · 2021-11-22
> **Response to Reviewer F2da**
>
> We sincerely thank the reviewer's constructive comments and valuable insights, which help improve the quality of our work significantly. We have carefully revised our paper according to your comments and suggestions. Please see our revised submission, where we have highlighted all major changes in **Red color**. Please also see our point-to-point responses as follows:
>
> > **Your Comment:** 1. The comparison of the results with (Zhang et al., 2018) may not be sufficiently fair. In particular, the assumptions made in that paper seems to be different from the current paper (e.g., Assumption 7 seems not to be needed in (Zhang et al., 2018). I would suggest the authors to make it clearer in Section 3.2 which assumptions are different from (Zhang et al., 2018), why they are essential, and how they help obtaining the improved results (like the non-asymptotic results) in the current paper. On a related point, the clarity of Section 3.2 also needs some improvement. For example, Assumption 2 is implied by Assumption 1, and so should not be stated as a separate assumption. Also, the notation of consensus weight matrix, $A_{\pi_{\theta}}$ and advantage functions all use A, which might cause some confusion.
>
> **Our Response:** Thanks for your constructive comments. In this revision, we have modified the original Assumption 2 to a lemma implied by Assumption 1 (see Lemma 1 on Page 12). Note that since we have changed the original Assumption 2 to a lemma as a result of Assumption 1 in this revision, the original Assumptions 3-7 become Assumptions 2-6 in this revision. To minimize the confusion, in the following response, we still use the old numbering 1-7 with the word "original" in front of the word "Assumptions."
>
> * Regarding assumptions in Section 3.2, we would like to clarify that the original Assumptions 6-7 in our paper are not used in (Zhang et al., 2018). The original Assumptions 6-7 are essential because of the following reasons: First, the original Assumption 6 ensures the optimal value function approximation is bounded in the critic step. This boundedness is crucial in the critic step analysis and further carried out into the actor step analysis. Second, the original Assumption 7 is necessary in the sense that, by the smoothness definition in Assumption 7, the upper bound on the stationary point can be characterized (cf. Eq.~(33) in the appendix). We also note that the paper (Zhang et al., 2018) assumes that the policy parameter is in a compact set, while we do not have such an assumption.
>
> * We also note that the reason we treat the paper (Zhang et al.2018) as a related work is that this paper also considered the average reward setting in MARL, which is the closest to our model in terms of the objective function. However, the key difference between our work and (Zhang et al. 2018) is that our work is the first to obtain finite-time convergence result under the local action observability constraint and in the average reward setting.
>
> * Regarding clarity of Section. 3.2, we have made the following changes in this revision:
>     * We have revised the original Assumption 2 as a lemma. Please see Lemma 1 on Page 12.
>     * In our original submission, the notation $A_{\pi_{\theta}}$ is the matrix defined in equation (1), and the consensus matrix is $A$.
>  To avoid confusion, in this revision, we have replaced the advantage function with the notation $\text{Adv}$. Thanks for your suggestions!
>
> > **Your Comment**: 2. The authors should also compare the results with the recent global optimality literature of actor-critic methods, such as [A,B]. Are there any essential challenges in obtaining global optimality bounds using the techniques introduced in papers such as [A,B]? Also, in [A], it is mentioned that from both theoretical and practical perspectives, bi-level actor-critic methods which first update the critic to sufficient accuracies as in the current paper are typically seldom adopted in practice. However, the numerical results in this paper seem to suggest that the bi-level/mini-batch updates might indeed be beneficial. Some more discussion and numerical experiments are needed to validate this (see the next point for more comments). I have some questions for the numerical experiments. The authors mentioned that constant step-sizes are key to the improvement over (Zhang et al., 2018). Then what if using the same constant step-sizes for the classical MARL algorithm in (Zhang et al., 2018)? And what about updating classical MARL in a mini-batch critic/bi-level manner as in this paper? I think providing these comparisons are important to understand what are essential in the modifications on top of the classical MARL algorithm. Otherwise, it gives me an impression that allowing for observing global actions is impeding the performance, which is counterintuitive.
>
> **Please continue to see our next response below.**

---

> > ### Author Response · Authors · 2021-11-22
> > **Response to Reviewer F2da (Continued)**
> >
> > **Our Response:** Thanks for your insightful comments. Due to the length of this comment, in what follows, we structure our response in several bullets for better readability.
> >
> > * *Comparisons to Works in the Global Optimality Literature:* In this paper, our focus is on characterizing the finite-time convergence rate to at least a stationary point. We focus on this performance metric because it is relevant to more general MARL settings and requires weaker assumptions compared to [A,B], where the over-parameterization setting and the special LQR problem structure are the key to the global optimality results. In addition, the problem setting in [A] is the discounted reward setting, as opposed to the average reward setting in our work. In the analysis in [A], the $\gamma$-contraction property is repeatedly utilized, where $\gamma$ is the discounting factor (see the first paragraph of Page 3 in [A] and Eq.~(5.4), and its interpretation on Page 13). The analysis in [B] is focused on the special LQR problem, whereas we studied the more general MARL setting. Although these works in the global optimality literature have different problem settings, we have cited these works in this revision and added the above discussions. We thank the reviewers for the pointers to these works.
> > We also agree that it is indeed very interesting to study the possibility of reaching global optimum as in [A,B]. This will be left as the next step in our future studies.
> >
> > * *Performance of Bi-level Actor-Critic Methods and Numerical Experiments:* We have added additional empirical comparison results with the variants of the classical MARL algorithm in (Zhang et al. 2018) as you suggested, including both constant step-sizes and batch sampling modifications. Our experiments showed that our algorithm outperforms the classical MARL algorithm and its variants with different system sizes. This suggests that, empirically, the double-loop structure combined with the consensus TD sharing approach might be beneficial. See Section A.2 for detailed comparisons.
> >
> > * *Numerical Result Comparisons to [Zhang et al. 2018]:* We note that the fact that observing joint action is not the mere factor that impedes the performance. Different from (Zhang et al., 2018), we have adopted a double-loop structure where in the TD learning subroutine (See Algorithm 1), where the value function approximation parameters are updated till convergence. In contrast, the value function approximation parameters in (Zhang et al., 2018) are updated only in a single step, which introduces larger errors in the actor step update.
> > These differences can potentially contribute to the better performance of our proposed algorithm.
> >
> > > **Your Comment:** 3. There are also some minor suggestions, as detailed below: In Assumption 5, why is $\Phi u\neq \mathbf{1}$ for any $u$ reasonable? I understand that this is a classical assumption, but still some explanations would be helpful.
> >
> > **Our Response:** Thanks for your question. The assumption $\Phi u \neq \mathbf{1}$ is reasonable. Note that, in practice, it often holds that $|\mathcal{S}| > K$, i.e., the size of the state space is greater than the feature mapping dimension. Thus, the matrix $\Phi\in \mathcal{R}^{|\mathcal{S}|\times K}$, $|\mathcal{S}|>K$ is a tall and skinny matrix. This implies that, mathematically, the linear equation system $\Phi u=\mathbf{1}$ is overdetermined. Thus, there is no solution under proper choice of the feature matrix $\Phi$. Moreover, this assumption ensures  $A_{\pi_{\theta}}$ defined in Eq.~(1) to be invertible (see the following reference [C]). This assumption in turn implies that the policy evaluation has a unique stable condition (please also see [Zhang et al., 2018]).
> >
> > > **Your Comment:** 4. Some more comments on what the quantity $A_{\pi_{\theta}}$ in (1) is would be helpful.
> >
> > **Our Response:** Thanks for the suggestion. In Eq. (1), $A_{\pi_{\theta}}$ is a matrix that characterizes the limiting point of TD(0) algorithm under the policy $\pi_{\theta}$. Specifically, the limiting point satisfies $A_{\pi_{\theta}} w^*_{\theta} + b_{\pi_{\theta}} = 0$, where $b_{\pi_{\theta}} = E_{s\sim d_{\theta}, a \sim \pi_{\theta}} [\phi(s)(\bar{r}(s,a)-J(\theta))]$ and $\bar{r}(s,a)=\frac{1}{N}\sum_{i\in\mathcal{N}}r^{i}(s,a)$. In the matrix form (Qiu et al., 2021), $A_{\pi_{\theta}}= \Phi^{T}D_{s}^{\theta}(P_s^{\theta}-I)\Phi$, where $P_s^{\theta}$ is the state transition matrix under policy $\pi_{\theta}$ and  $D_s^{\theta} = \text{Diag}(d_{\theta}(s_1),\cdots,d_{\theta}(s_{|\mathcal{S}|}))$ is the diagonal distribution matrix.
> >
> > > **Your Comment:** 5. On page 6, 2) The Actor Step, “where in our model” might better be “whereas in our model”.
> > On page 7, what is “the weight information of other agents”?
> >
> > **Please continue to see our next response below.**

---

> > > ### Author Response · Authors · 2021-11-22
> > > **Response to Reviewer F2da (Continued)**
> > >
> > > **Our Response:** Thanks for your suggestions, we have rephrased the sentence on Page 6 as you suggested.
> > > The sentence "the weight information of other agents" on Page 7 means that, for a given matrix $W$, agent $i$ only needs to know the weight $W_{ij}$, where $j$ is a neighbor of $i$ and agent $i$ does not need the information $W_{mn}$ where $m\neq i$.
> > >
> > > > **Your Comment:** 6. It would be helpful to refer to the x-axis in the plots as “number of samples” instead of “sample complexity”. And just to double check, does one sample mean tuple of $(s,a,r,s')$?
> > >
> > > **Our Response:** Thanks for your suggestion. We have made the change in this revision as you suggested. And yes, "one sample" here means the tuple of $(s,a,r,s')$.
> > >
> > > [A] Fu, Zuyue, Zhuoran Yang, and Zhaoran Wang. ``Single-timescale actor-critic provably finds globally optimal policy." arXiv preprint arXiv:2008.00483 (2020).
> > >
> > >
> > > [B] Yang, Zhuoran, Yongxin Chen, Mingyi Hong, and Zhaoran Wang. ``On the global convergence of actor-critic: A case for linear quadratic regulator with ergodic cost." arXiv preprint arXiv:1907.06246 (2019).
> > >
> > > [C] Wu, Y., Zhang, W., Xu, P. and Gu, Q., 2020. ``A finite time analysis of two time-scale actor critic methods.'' arXiv preprint arXiv:2005.01350.

---

> > > > ### Comment · Reviewer_F2da · 2021-12-06
> > > > **Thanks for the revision and response!**
> > > >
> > > > I would like to thank the authors for the careful revision, detailed response and informative explanations. My major concerns have all been clearly addressed. In particular, the additional comparisons with (Zhang et al., 2018) and the global optimality literature are clear and the additional numerical experiments and explanations validating the benefits of bi-level actor-critic (AC) methods are both convincing and insightful. I have increased my scores accordingly. I encourage the authors to mention the findings that bi-level AC might indeed perform better than single-loop AC in practice (at least for the average reward multi-agent setting in this paper) and provide their explanations in the early parts of the paper (like the abstract and introduction), as this might invoke broader future discussions and investigations in the community.

---

### Official Review · Reviewer_v83u · 2021-11-07

**Correctness:** 3
**Technical Novelty And Significance:** 3
**Empirical Novelty And Significance:** 2
**Recommendation:** 6
**Confidence:** 4

**Main Review:**

This paper is well written and easy to follow in general. Its theoretical contribution on the first finite time convergence result (to the stationary point) for fully decentralized MARL in the average reward setting is important. However, I have some comments and questions for the authors, which are listed below.
1. Why is the work (Chen et al., 2021) independent of the number of agents? Is it because of the difference between discounted setting and the average reward setting?
2. Assumption 2 should be a lemma or a proposition under Assumption 1. Also, I think Assumption 7 should be able to be replaced with some more explicit conditions (see for example Lemma 10 in [1]). Also it would be better if the authors can comment on how essential this assumption is as it is not needed in (Zhang et al., 2018).
3. Please better highlight the differences from (Zhang et al., 2018) in Algorithm 2, and which parts are improvements that lead to better theoretical guarantees and numerical results and which parts are sacrifices to allow for private (instead of joint) actions observability. In particular, in Section 4, the explanation on the modification of the actor step to allow for private action observations is unclear. It's a bit hard to follow what's the relationship among the global, networked and local TD errors and what are changed compared to the TD error computation in (Zhang et al., 2018). It would also be helpful to comment more which changes are to allow for the more challenging private action observation assumption, and which changes (also) contribute to some theoretical and numerical improvements (as suggested in the subsequent sections).
4. At the end of Section 5, it seems that the comment on the $O(\epsilon^{-1})$ sampling does not necessarily indicate that (Zhang et al., 2018)'s overall communication complexity is empirically worse (with appropriate hyper-parameter choices, such as step-sizes and mini-batches). In fact, synchronous actor and critic updates (instead of first updating the critic for many steps and then updating the actor) is more frequently adopted in practice.
5. It looks a bit weird to me why observing the joint actions does not seem to have any benefits according to the numerical experiments. If I understand correctly, in addition to addressing the private action observation challenge, this paper also considers mini-batch updates and constant step-sizes modifications. These can also be easily applied to the classical MARL algorithm in (Zhang et al., 2018). For a fair empirical comparison, I think it's important to provide some comparisons with the mini-batch constant step-size version of the baseline classical MARL algorithm.

References:
[1] X. Guo, A. Hu and J. Zhang. Theoretical Guarantees of Fictitious Discount Algorithms for Episodic Reinforcement Learning and Global Convergence of Policy Gradient Methods.

**Summary Of The Paper:**

This paper studies the cooperative average reward fully decentralized multi-agent reinforcement learning (MARL) problems, where the agents interact with their neighbors over a communication network. It proposes a consensus-based actor-critic algorithm and shows its convergence to the stationary point. The convergence rate and sample complexity of this algorithm are provided and comparison with existing algorithm is shown in the numerical experiments.

**Summary Of The Review:**

This paper proposes a new actor-critic algorithm to deal with a cooperative average reward fully decentralized MARL and provides the first finite time convergence result. The theoretical results are important. However, the differences with existing literature are not well explained and numerical experiments need some improvements.

---

> ### Author Response · Authors · 2021-11-22
> **Response to Reviewer v83u**
>
> We sincerely thank the reviewer's constructive comments and valuable insights, which help improve the quality of our work significantly. We have carefully revised our paper according to your comments and suggestions. Please see our revised submission, where we have highlighted all major changes in **Red color**. Please also see our point-to-point responses as follows:
>
> > **Your Comment:** 1. Why is the work (Chen et al., 2021) independent of the number of agents? Is it because of the difference between discounted setting and the average reward setting?
>
> **Our Response:** Thanks for your comments. Please see our response to each of your questions below:
>
> * In the result of (Chen et al., 2021), the number of agent N is treated as a hidden constant in the Big-O notation, rather than being explicitly expressed as a parameter like in our work. To see this, note that the overall convergence rate of the decentralized Actor-Critic(DAC) in (Chen et al., 2021) to a stationary point can be found in Theorem 1 of (Chen et al., 2021), where constants $c_4, c_5, c_6, c_8$ are all $M$-dependent, where $M$ is the number of agents in that paper (same as $N$ in our work). The precise dependency of these constants on $M$ in DAC is summarized on Page 39 of their appendix. Due to this $M$-dependence of the convergence rate, their sample complexity result is also agent-size-dependent.
>
> * Although the order-wise dependence relationships on the number of agents $N$ are different in our paper and the work in (Chen et al., 2021), this difference is not due to the difference between the average reward setting and discounted reward setting. The order-wise difference in system size dependence arises because of the different approaches in information sharing.
> In the paper (Chen et al., 2021), the agents share noisy rewards, whereas, in our paper, we share the TD errors.
> In our analysis, sharing TD errors involves consensus errors of the value function parameters, which brings in error term scaled with system parameter $N$, see Eq. (35). This error in terms of agent size is possible to be minimized order-wise by sharing the noisy sum of reward and local long term average reward estimation, i.e. $r^{i}_t+\mu^{i}_t$.
> A key merit of our approach is that we do not need a reward resampling process for each agent at each sample.
>
> > **Your Commment:** 2. Assumption 2 should be a lemma or a proposition under Assumption 1. Also, I think Assumption 7 should be able to be replaced with some more explicit conditions (see for example Lemma 10 in [1]). Also it would be better if the authors can comment on how essential this assumption is as it is not needed in (Zhang et al., 2018).
>
> **Our Response:** Thanks for your insightful comments and related references. According to your comments, we have made the following revisions:
> * We have revised Assumption 2 as a lemma implied by Assumption 1. See Lemma 1 on Page 12.
> * We agree that we can replace the original Assumption 7 by fixing the policy as the class of soft-max policy and combining such a policy class with the condition in Assumption 1, where the original Assumption 7 holds as a result as in [1].
> * Assumption 7 is important because by the smoothness definition in Assumption 7, the upper bound on the stationary point can be characterized (see Eq. (33) in the Appendix). With this stronger assumption, we are able to obtain the finite-time convergence result, whereas in (Zhang et al,. 2018), the convergence is only asymptotic.
>
> > **Your Comment:** 3. Please better highlight the differences from (Zhang et al., 2018) in Algorithm 2, and which parts are improvements that lead to better theoretical guarantees and numerical results and which parts are sacrifices to allow for private (instead of joint) actions observability. In particular, in Section 4, the explanation on the modification of the actor step to allow for private action observations is unclear. It's a bit hard to follow what's the relationship among the global, networked and local TD errors and what are changed compared to the TD error computation in (Zhang et al., 2018). It would also be helpful to comment more which changes are to allow for the more challenging private action observation assumption, and which changes (also) contribute to some theoretical and numerical improvements (as suggested in the subsequent sections).
>
> **Our Response:** Thanks for your comments. For better readability, we would like to clarify or answer each sub-question in the following bullets:
>
> **Please continue to see our next response below.**

---

> > ### Author Response · Authors · 2021-11-22
> > **Response to Reviewer v83u (Continued)**
> >
> > * *Algorithmic Differences from (Zhang et al., 2018):* In this paper, the major difference compared to (Zhang et al. 2018) is that, due to the lack of joint action knowledge, we rely on consensus-based techniques and a double-loop structure to exchange the local TD error information, which gradually drives the computation of the global TD error to the average of the local TD errors. (Please see the last set of equations on Page 6). In comparison, due to the availability of joint action information, there is no need to perform consensus in Algorithm 2 of (Zhang et al., 2018).
> >
> > * *Performance Improvements Compared with (Zhang et al., 2018):* We would like to clarify that, due to fundamental differences in the settings and assumptions, the performance of our work and (Zhang et al., 2018) is *not* directly comparable.
> > Our work is related to (Zhang et al., 2018) because both papers considered the average reward setting in MARL.
> > However, our work differs from (Zhang et al., 2018) in the following key aspects: i) We assume that actions are only locally observable under some different assumptions (i.e. the original Assumption 6 and 7), rather than the joint action observations as in (Zhang et., al 2018).
> > For example, the ratio of sample number over the number of communication round is $O(\epsilon^{-1})$, where as in (Zhang et al., 2018) this ratio is 1. This ratio implies that we have better communication efficiency.
> >
> > * *Sacrifice for allowing private action observability:* The major sacrifice for allowing private action observability is the increased iteration rounds of communication, i.e. $t_\text{gossip}$, to reach a good estimate of the global TD error. However, the consensus-based approach allows private action observability, and the communication complexity is compensated via a batch sample (see Line 12-17 in Algorithm 2).
> >
> > * We would like to further clarify the relationship among the global, networked, and local TD errors. The global TD error $\delta$ is the average of the local TD errors $\delta_i$. Let $\delta_{\text{local}}=(\delta_1,\cdots,\delta_N)^{T}$. Under the doubly stochastic weight matrix $A$, we have $\delta=\lim_{t\to\infty} A^{t}\delta_{\text{local}}$. It is this consensus operation allows the global TD error being estimated by the local TD errors.
> >
> > > **Your Comment:** 4. At the end of Section 5, it seems that the comment on the $O(\epsilon^{-1})$ sampling does not necessarily indicate that (Zhang et al., 2018) overall communication complexity is empirically worse (with appropriate hyper-parameter choices, such as stepsizes and mini batches). In fact, synchronous actor and critic updates (instead of first updating the critic for many steps and then updating the actor) is more frequently adopted in practice.
> >
> > **Our Response:** Thank you for the comment. We agree that the communication complexity in (Zhang et al., 2018) can indeed be improved by the batch sampling approach. However, we emphasize that, from the theoretical perspective, our approach provides the finite-time convergence results, whereas in [Zhang et al. 2018], the convergence is only asymptotic (i.e., there is no convergence rate result in [Zhang et al. 2018]). From the practical perspective, when given a precision parameter $\epsilon>0$ to the stationary point, our theoretical results can provide practical guidelines on how to choose the batch sizes for both critic step and actor step. Moreover, empirically, the numerical results in Figure 2 (a)-(c) (in Section A.2 on Page 13) show that even with a constant step-size or batch sampling modification or both, our TD-sharing algorithm still outperforms the classical MARL algorithm and its variants in various cases with different number of agents, verifying our contributions of performing finite-time analysis instead of an asymptotic analysis.
> >
> > The reason that we prefer a double-loop structure is that it enjoys more tractability in theoretical performance analysis. By using such a double-loop structure, we can decouple the analysis of the critic and actor steps and analyze these two steps alternatively. Specifically, given the current policy, we can obtain a good estimate of the value functions of the policy and control the error by adjusting the iteration times and batch size. Once the critic step is finished, we can use the estimated value functions to update the policy parameter in the actor step. But we thank the reviewer's question/suggestion on synchronous actor and critic updates, which will be an important topic in our future studies.
> >
> > **Please continue to see our next response below.**

---

> > > ### Author Response · Authors · 2021-11-22
> > > **Response to Reviewer v83u (Continued)**
> > >
> > > > **Your Comment:** 5. It looks a bit weird to me why observing the joint actions does not seem to have any benefits according to the numerical experiments. If I understand correctly, in addition to addressing the private action observation challenge, this paper also considers mini-batch updates and constant step-sizes modifications. These can also be easily applied to the classical MARL algorithm in (Zhang et al., 2018). For a fair empirical comparison, I think it's important to provide some comparisons with the mini-batch constant step-size version of the baseline classical MARL algorithm.
> > >
> > > **Our Response:** Thank you for the suggestion. In this revision, we have added additional empirical comparison results with modified classical MARL algorithms. Specifically, we have modified the classical MARL algorithm into variants: i) constant step-size MARL, ii) batch MARL, and iii) batch constant step-size MARL algorithms. We have conducted simulations to empirically show the effects of constant step-sizes and batch sampling on the classical MARL algorithm. The simulation results show that our algorithm outperforms the classical MARL algorithm and its variants. Also, the constant step-sizes and batch sampling does improve the performance of classical MARL algorithm to some extent, especially when system size is larger (specifically $N=10$ and $N=15$). In addition, we have also included the comparison of our algorithm in the average setting versus the discounted counterpart and the network structure on the performance of our algorithm. See Section A.2 for a detailed comparisons.
> > >
> > > References: [1] X. Guo, A. Hu and J. Zhang. Theoretical Guarantees of Fictitious Discount Algorithms for Episodic Reinforcement Learning and Global Convergence of Policy Gradient Methods.

---

> > > > ### Comment · Reviewer_v83u · 2021-12-06
> > > > **response to rebuttal**
> > > >
> > > > The authors provide nice explanations on my concerns and I have revised the score.

---

### Decision · Program_Chairs · 2022-01-20

**Decision:**

Accept (Spotlight)

**Comment:**

This paper provides actor-critic method for fully decentralized MARL. The results remove some of the restrictions from existing results and have also obtained a sample bound that matches with the bound in single agent RL. The authors also give detailed responses to the reviewers' concerns. The overall opinions from the reviewers are positive.